# Physically Constrained Covariance Inflation from Location Uncertainty

Yicun Zhen[1], Valentin Resseguier[2,3], and Bertrand Chapron[4]

[1]College of Oceanography, Hohai University, Nanjing, China
[2]LAB SCALIAN DS, Rennes, France
[3]INRAE, OPAALE, Rennes, France
[4]Laboratoire d'Océanographie Physique et Spatiale, Ifremer, Plouzaé, France

**Correspondence:** Yicun Zhen (zhenyicun@proton.me)

**Abstract.** Motivated by the concept of "location uncertainty", initially introduced in Mémin (2014), a scheme is sought to perturb the "location" of a state variable at every forecast time step. Further considering Brenier's theorem Brenier (1991), asserting that the difference of two positive density fields on the same domain can be represented by a transportation map, we demonstrate that the perturbations consistently define an SPDE from the original PDE. It ensues that certain quantities, up to the user, are conserved at every time step. Remarkably, derivations following both the SALT Holm (2015) and LU Mémin (2014); Resseguier et al. (2017a) settings, can be recovered from this perturbation scheme. Still, it opens broader applicability since it does not explicitly rely on Lagrangian mechanics or Newton's laws of force. For illustration, a stochastic version of the thermal shallow water equation is presented.

## 1 Introduction

Data assimilation is meant to extract information from measurements to improve the state estimate. Kalman-filter-based and particle-filter-based methods are now commonly used for academic studies and operational forecasts. For both methods, the estimate of state variable and the uncertainty quantification of the estimate of a state variable are repeated at each data assimilation cycle. In the classical Kalman filter, this uncertainty is represented by a covariance matrix. In Monte-Carlo-based methods (i.e. the ensemble Kalman filters and particle filters, etc.), it is represented by the spread of the ensemble members or particles. The uncertainty of the state estimate is further part of the input for the next data assimilation cycle. Frequently observed, the uncertainty can be underestimated in nonlinear numerical experiments when there is no model noise Schlee et al. (1966); Harlim and Majda (2010); Franzke et al. (2015). As a consequence, the state estimate in the subsequent time steps may not be efficiently adjusted by the physical measurements: the system is over-confident about its current state estimate. This phenomenon is usually referred to as filter divergence, possibly associated to the "curse of dimensionality" (see for instance Daum and Huang (2003)).

To address the latter issue, "covariance localization" has been developed for both Kalman-filter-based methods and particle filters Houtekamer and Mitchell (2001); Poterjoy (2016). To further mitigate filter divergence, a practical strategy is to inflate the uncertainty estimate at each forecast time step or each data assimilation cycle Anderson (2007); Tibshirani and Knight

(1999); Li et al. (2009); Kotsuki et al. (2017); Ying and Zhang (2015); Miyoshi (2011); Raanes et al. (2019); Zhen and Harlim (2015). For geophysical applications, the uncertainty is then often inflated by rescaling the ensemble covariance in order to match bias and variance. A natural alternative is the addition of noise in the dynamical equations.

In the context of ensemble/particle-based methods, the uncertainty is usually inflated by artificially perturbing each ensemble member/particle. We refer the reader to Resseguier et al. (2021) for a review on the subject. It is then a natural question to ask: is there a mathematical principle to guide this uncertainty inflation? In the fluid dynamics community, random forcings are not introduced for inflation, but to mimic the intermittent back-scattering of energy from small scales toward large scales. Among those approaches, we may mention the stochastic Lagrangian models Pope (1994) and the Eulerian Gaussian backscatterings of EDQNM Orszag (1970); Leith (1971). Additive noise models, like the linear inverse models Penland and Sardeshmukh (1995), have then also been proposed for filtering purposes, and thoroughly reviewed by Tandeo et al. (2020). Most methods mainly focus on comparing the estimated uncertainty and the statistics of the innovation process, but ignore other mathematical/physical aspects (for instance, the conservation laws, etc.). Other empirical approaches, referred to as SPPT Buizza et al. (1999) and SKEBS Berner et al. (2009), introduce multiplicative noise, with success in operational weather and climate forecast centers Franzke et al. (2015). Still many drawbacks have been reported, above all violations of conservation laws Reynolds et al. (2016); Leutbechner et al. (2016). Recently, the operational ocean circulation model NEMO has also been randomized (e.g. Leroux et al., 2022), but again, without conservative considerations.

Several authors proposed schemes specifically to enforce energy conservation or at least a given energy budget (e.g. Sapsis and Majda, 2013; Gugole and Franzke, 219; Resseguier et al., 2021). To better constrain non-Gaussian schemes, many authors rely on physics and possibly on time-scale separation. Introduced by Hasselmann (1976), it is generally associated with the rigorous theories of averaging and homogenization. Majda et al. (1999) decomposed the state variable into slow-varying modes $x_j$ and fast-varying modes $y_j$. The authors demonstrated that the interaction term between $x_j$ and $y_j$, in the equation for $x_j$, can be modeled as a stochastic process solely in terms of $x_j$'s, as the ratio of the time scales of $x_j$ and $y_j$ tends to 0. Nevertheless, homogenization methods, like Majda et al. (1999), may also lead to violation of energy conservation, even though some workarounds exist Frank and Gottwald (2013); Jain et al. (2014).

In Brzeźniak et al. (1991), later modified in Mikulevicius and Rozovskii (2004); Flandoli (2011) and Mémin (2014); Resseguier et al. (2017a, 2021), preservation of kinetic energy is specifically emphasized. The true velocity of an incompressible flow is decomposed into a regular component and a turbulent one, and the latter modeled by a stochastic noise. Mikulevicius and Rozovskii (2004) and Mémin (2014) further derived stochastic Navier-Stokes equations. For these two approaches, the large-scale advecting velocity differs, induced by different regularisation of the Newton' second law. Following another path, considering the Hamilton's principle with a stochastic advection constraint on Lagrangian fluid trajectories, Holm (2015) also proposed a consistent stochastic setting, i.e stochastic advection by Lie transport (SALT). In particular, this derivation preserves Kelvin's circulation. Similarities and differences between these different stochastic frameworks are discussed in Resseguier et al. (2020).

From another perspective, the classical optimal transport theory suggests that the difference of two smooth positive density fields ($\rho_1$ and $\rho_2$) on a bounded domain $\Omega$ can be described by a transportation map: $T : \Omega \to \Omega$. More specifically, there exists

a diffeomorphism $T$ of $\Omega$ to transform $\rho_1$ to $\rho_2$ under the diffeomorphism $T$ with a minimal cost. Broadly speaking, $T$ can be interpreted as how much $\rho_2$ differs from $\rho_1$, and $T$ operates as a location correction. Indeed, starting from the same initial condition $\rho(t)$, suppose that $\rho_1 = \rho^{\text{model}}(t + \Delta t)$ is the model forecast and $\rho_2 = \rho(t + \Delta t)$ is the true forecast. The additional uncertainty of $\rho_1$ due to model error can then be represented by a random $T$. It further suggests that the inflation of uncertainty can be achieved by casting a random $T$ on each ensemble member/particle.

Motivated by such an optimal transport perspective and the concept of "location uncertainty", proposed in Mémin (2014), a new strategy can thus seek to design a well constrained "location perturbation" of the state variable. Specifically, the idea of covariance inflation can be informally generalized to physical fields that are not always positive, i.e. physical fields other than the density field. Mathematically, a density field $\rho$ is naturally associated to a differential $n$-form $\theta_\rho$, where $n = \dim \Omega$. The statement "$\rho_1$ transforms to $\rho_2$ under the diffeomorphism $T$" is equivalent to the mathematical relation $\theta_{\rho_1} = T^* \theta_{\rho_2}$, where $T^*$, acting on all differential forms, is the pull-back operator induced by $T$, or equivalently, $\theta_{\rho_2} = (T^{-1})^* \theta_{\rho_1}$. Therefore, a random $T$ (or equivalently, $T^{-1}$) could induce a perturbation of any differential $k-$form.

To implement a physically-constrained perturbation scheme, the state variable $S$ under consideration must then be associated to some differential form $\theta$, i.e. construct a 1-1 correspondence between snapshots of $S$ and snapshots of $\theta$. Note, this can be generalized to other types of tensor fields. It will be demonstrated (section 5) that it is indeed sometimes helpful to choose $\theta$ to be a contravariant tensor field other than differential forms. Yet, it must be stressed that associating the state variable $S$ to a differential form $\theta$ is a key important step.

Correspondingly, at each forecast time step, the covariance inflation should follow 4 steps:

- Step 1, find $\theta(t)$ based on $S(t)$.

- Step 2, construct a random diffeomorphism $T : \Omega \to \Omega$.

- Step 3, replace $\theta(t)$ with $T^* \theta(t)$ and calculate $S(t)$ based on the new value of $\theta(t)$.

- Step 4, calculate the forecast $S(t + \Delta t)$ based on the new value of $S(t)$.

Associating $S$ to different $\theta$ shall then be constrained by different conservation laws for the perturbation scheme. More precisely, certain physical quantities are conserved in step 3, no matter how $T$ is constructed or realized in step 2. We emphasize that the conservation law of the perturbation scheme merely depends on the choice of $\theta$, but is independent of the dynamics of the original deterministic system. A resulting SPDE will conserve a given quantity only if both the perturbation scheme and the original deterministic system conserve that quantity. We also remark that this scheme can not conserve all the physical quantities at the same time unless additional constraints upon the parameters are imposed. Hence the users must choose by themselves which physical quantity to conserve.

In summary, the key perspective of this manuscript is that the displacement vector field of physical state variables should be determined by the tensor fields associated to the physical fields. The advantage of this perspective is that certain physical quantities can be conserved while applying a displacement vector field to transfer the original physical field. A direct application of this perspective is the physically constrained covariance inflation scheme proposed in this manuscript. When the tensor

fields are positive $n-$forms on a bounded domain that have the same total mass, Brenier's theorem shows that the 'optimal' displacement vector field exists and is unique, for a given cost function. In this case, the optimality of displacement vector field is well-defined. In other cases, the issue of 'optimality' together with the existence and uniqueness of 'optimal' displacement vector field need to be carefully explored. We reserve this to future study.

This paper is organized as follows. Section 2 is a brief introduction of optimal transport theory. In section 3 we present the perturbation scheme in detail, including the motivation, the specific techniques in derivation, and several examples. In section 4, the resulting perturbation scheme is then compared with the stochastic advection by Lie transport (SALT) equations Holm (2015) and the location uncertainty (LU) equations Mémin (2014). For properly chosen $\theta$ and $T_t$, it is demonstrated that both SALT and LU settings are recovered within the proposed framework. To illustrate our purpose, a stochastic version of the thermal shallow water equation is then derived in section 5. A final conclusion and discussion is given in section 6.

**Convention of notation:**

- The letter $i$ only refer to the $i-$th independent Brownian motion. The letters $p, q, j, k$ refer to the components if $p, q, j, k$ are upper indices.

- Einstein's convention on summation (applies to all indices except $i, j$): if $p$ is shown in both upper and lower indices, then the summation over $p$ automatically applies.

- Summation over $i, j, p$ automatically applies in all equations. For instance, $e_i$ refers to $\sum_i e_i$, and $y_j$ refers to $\sum_j y_j$

## 2 Monge's formulation of optimal transport problem and Brenier's answer

Hereafter we briefly summarize some necessary concepts and results in optimal transport theory. Let $\Omega$ be a bounded domain in a $n-$dimensional Euclidean space.

**Definition 2.0.1** (Monge's optimal transport problem). *Given cost function $c(x, y) \geq 0$ and probability measures $\mu, \nu \in \mathcal{P}(\Omega)$,*

$$minimize\ \mathbb{M}(T) = \int_\Omega c(x, T(x))d\mu(x) \tag{1}$$

*over $\mu$ measurable maps $T : \Omega \to \Omega$ subject to $\nu = T_{\#}\mu$.*

Here the probability measures $\mu$ and $\nu$ are interpreted as mass distributions with total mass equal to 1. The map $T$ is called a transport plan which moves the mass $d\mu(x)$ at location $x$ to location $T(x)$, with the cost $c(x, T(x))$ per unit of mass. Therefore the quantity $\mathbb{M}(T)$ is the total cost of the transport plan $T$. The constraint $\nu = T_{\#}\mu$ is interpreted as that $T$ transports the mass distribution $\mu$ to the mass distribution $\nu$. In the case that $T$ is a diffeomorphism and that both $\nu$ and $\mu$ have smooth densities, i.e. assume that $d\nu(x) = f(x)d^n x$ and $d\mu(x) = g(x)d^n x$ for some smooth functions $f, g$ on $\Omega$,

$$\nu = T_{\#}\mu \iff g(x) = f(T(x))|J_T(x)|, \tag{2}$$

where $J_T(x)$ refers to the Jacobian matrix of $T$ at $x$. If we associate $\nu$ and $\mu$ to differential $n-$forms $\theta_\nu = f dx^1 \wedge \cdots \wedge dx^n$ and $\theta_\mu = g dx^1 \wedge \cdots \wedge dx^n$, then

$$\nu = T_\# \mu \iff \theta_\mu = T^* \theta_\nu. \tag{3}$$

Brenier Brenier (1991) proved the existence and uniqueness of the solution to the Monge's optimal transport problem for $c(x,y) = |x-y|^2$. To better illustrate how optimal transport theory motivates us, we consider the following simplified version of Brenier's theorem.

**Theorem 2.1** (Brenier, simplified version). *Let $\mu$ and $\nu$ be measures with bounded smooth density on a bounded domain $\Omega \subset \mathbb{R}^n$. Let $c(x,y) = |x-y|^2$. Then there is a convex function $\phi : \Omega \to \mathbb{R}$, such that $(\nabla \phi)_\# \mu = \nu$. And $\nabla \phi : x \to x + \nabla \phi|_x$, defined $\mu-$almost everywhere, is the unique solution to the Monge's optimal transport problem.*

The convexity of $\phi$ implies that the map $\nabla \phi$ is one-to-one. Broadly speaking, Brenier's theorem implies that the difference of two density fields can be represented by a transportation map $T$.

## 3 The Perturbation Scheme

Consider a compressible flow on a bounded domain $\Omega$. Let $\rho$ denote the density field. Let $\rho^{\text{model}}(t + \Delta t)$ and $\rho^{\text{true}}(t + \Delta t)$ be the model forecast and the true forecast starting from the same density field at time $t$. If we assume that the model forecast and the truth have the same total mass, Brenier's theorem says that there exists a diffeomorphism $T : \Omega \to \Omega$ so that

$$\rho^{\text{true}}(x, t + \Delta t) = \rho^{\text{model}}(T(x), t + \Delta t) J_T(x). \tag{4}$$

Note that the transportation $T$ hereinafter is equivalent to the mapping $T^{-1}$ used in the introduction. Eq.(4) can further be written in terms of differential form. Let $\theta_\rho = \rho dx^1 \wedge \ldots \wedge dx^n$, then Eq.(4) is equivalent to

$$T^* \theta_\rho^{\text{model}}(t + \Delta t) = \theta_\rho^{\text{true}}(t + \Delta t). \tag{5}$$

For general differential forms $\theta$, it is unclear whether a diffeomorphism $T$ always exists that satisfies Eq.(5). However, Eq.(5) provides us with a tool for covariance inflation by constructing a random $T$ at every infinitesimal time step. At each time step we construct a small perturbation $T$:

$$T_t(x) = x + a(t,x)\Delta t + e_i(t,x)\Delta \eta_i(t), \tag{6}$$

where $a(t,x), e_i(t,x) \in \mathbb{R}^n$, $\Delta \eta_i(t) \sim \mathcal{N}(0, \Delta t)$ is a random number. Essentially, $T_t(x) - x$ can be interpreted as a "location error" caused by the model error. In Eq.(6), $a(t,x)\Delta t$ refers to a systematic location error, and $e_i \Delta \eta_i$ refers to a random location error. Stated in the introduction, the state variable $S$ must first be associated to a differential form $\theta$. Then at every time step, $T_t$ induces a perturbation of $\theta(t)$ by $\theta(t) \to T_t^* \theta(t)$. It hence induces a perturbation of the state variable $S(t)$. A forecast is then performed based on the perturbed state. Consequently, this perturbation scheme derives an SPDE from the original PDE.

This procedure can also be generalized to other types of tensor fields. We refer to Chern et al. (1999) for a rigorous definition of the tensor fields and the wedge algebra. For instance, we may choose $\theta = \rho \frac{\partial}{\partial x^1} \wedge \cdots \wedge \frac{\partial}{\partial x^n}$, where $\{\frac{\partial}{\partial x^i}\}_{i \leq n}$ forms a global basis of the tangent field. Then $T_t$ induces a perturbation of $\theta$ by $\theta(t) \to T_{t*}\theta$, where $T_{t*}$ is the push-forward operator induced by $T_t$. In section 5, such a generalization is found useful in the example of thermal shallow water equation.

**Remark 1.** *When $\theta$ is a mixture of covariant and contravariant tensor fields, the perturbation scheme is slightly more complicated. Assume that $T_t : \Omega_1 \to \Omega_2$ is a diffeomorphism, and $\theta = v \otimes \omega$ where $v$ and $\omega$ are contravariant or covariant tensor fields respectively on $\Omega_2$. Then $T_t^* \omega$ is a covariant tensor field on $\Omega_1$. However, $T_t$ can not directly induce a contravariant tensor field on $\Omega_1$. In order to get a tensor field on $\Omega_1$, we consider $T_t^{-1} : \Omega_2 \to \Omega_1$, and apply the push-forward operator on $v$. In sum, we may define the perturbation to be*

$$\theta(t) \to \left((T_t^{-1})_* v\right) \otimes \left(T_t^* \omega\right). \tag{7}$$

*Appendix A derives the expression of $T_t^{-1}$ directly from the expression of $T_t$.*

## 3.1 Calculation of $T_t^* \theta$ (or $T_{t*}\theta$)

A rigorous mathematical definition and calculation of $T_t$ and $T_t^*$ should be given in terms of stochastic flows of diffeomorphisms and its Lie derivatives. A brief discussion of the relationship between $T_t^*$ and the Lie derivative is given in section 4.1. We further refer to Leon (2021) a detailed definition of the Lie derivative. Yet, to rapidly assess $T_t^* \theta$ (or $T_{t*}\theta$), a Taylor expansion and Itô's lemma can be used.

Given coordinates $(x^1, ..., x^n)$, when $\theta$ is a differential $k-$form, it can be written as

$$\theta = \sum_{i_1 < ... < i_k} f^{i_1, ..., i_k} dx^{i_1} \wedge \cdots \wedge dx^{i_k}. \tag{8}$$

Then

$$T_t^* \theta = \sum_{i_1 < ... < i_k} f^{i_1, ..., i_k}(T_t(x)) T_t^* (dx^{i_1} \wedge \cdots \wedge dx^{i_k}). \tag{9}$$

Given in appendix B, a Taylor expansion and Itô's lemma are applied to expand $T_t^* \theta$, leading to compactly write

$$T_t^* \theta = \theta + \mathcal{M}(\theta)\Delta t + \mathcal{N}_i(\theta)\Delta \eta_i, \tag{10}$$

for some differential $k-$forms $\mathcal{M}(\theta)$ and $\mathcal{N}_i(\theta)$. Hereafter, several examples of $T_t^* \theta$ are presented.

The full derivation of these examples are skipped. We further express all the terms in coordinates. For instance, we replace $\langle \nabla f, a \rangle$ with $a^j \partial_{x^j} f$, where, by convention of notation, $a^j \partial_{x^j} f = \sum_j a^j \frac{\partial f}{\partial x^j}$. Similarly, $e_i^\top H_f e_i$ is replaced with $e_i^p e_i^q \partial_{x^p} \partial_{x^q} f$.

**Remark 2.** *When $\theta = f \frac{\partial}{\partial x^{i_1}} \wedge \cdots \wedge \frac{\partial}{\partial x^{i_k}}$ is a contravariant tensor field,*

$$T_{t*}\theta = f(T_t^{-1}(x)) T_{t*} (\frac{\partial}{\partial x^{i_1}} \wedge \cdots \wedge \frac{\partial}{\partial x^{i_k}}). \tag{11}$$

*The formula for $T_t^{-1}$ is derived in appendix A. Then the expression of $f(T_t^{-1}(x))$, $T_{t*} \frac{\partial}{\partial x^{i_1}} \wedge \cdots \wedge \frac{\partial}{\partial x^{i_k}}$ and $T_{t*}\theta$ can be derived step by step in a similar way as in appendix B.*

**Example 3.1.1.** When $\theta = f$ is a function (differential $0-$form),

$$(T_t^* \theta) = f + \left(a^j \partial_{x^j} f + \tfrac{1}{2} e_i^p e_i^q \partial_{x^p} \partial_{x^q} f\right) \Delta t + e_i^p \partial_{x^p} f \Delta \eta_i \tag{12}$$

**Example 3.1.2.** When $\theta = dx^1 \wedge dx^2 \wedge \cdots \wedge dx^n$,

$$T_t^* \theta = \left\{ 1 + \left(\partial_{x^p} a^p + \tfrac{1}{2} J_i\right) \Delta t + \partial_{x^p} e_i^p \Delta \eta_i \right\} \theta, \tag{13}$$

where $J_i = \partial_{x^p} e_i^p \partial_{x^q} e_i^q - \partial_{x^p} e_i^q \partial_{x^q} e_i^p$.

**Example 3.1.3.** When $\theta = f dx^1 \wedge \cdots \wedge dx^n$,

$$
\begin{aligned}
T_t^* \theta = \Big\{ & f + \Big((\partial_{x^p} a^p + \tfrac{1}{2} J_i) f + (a^p + e_i^p \partial_{x^q} e_i^q) \partial_{x^p} f + \tfrac{1}{2} e_i^p e_i^q \partial_{x^p} \partial_{x^q} f\Big) \Delta t \\
& + (\partial_{x^p} e_i^p f + e_i^p \partial_{x^p} f) \Delta \eta_i \Big\} dx^1 \wedge \cdots \wedge dx^n
\end{aligned} \tag{14}
$$

**Example 3.1.4.** When $\theta = f^j dx^j$ (note that by the convention of notation, $f^j dx^j = \sum_{j=1}^n f^j dx^j$),

$$
\begin{aligned}
T_t^* \theta = \Big\{ & f^j + (a^p \partial_{x^p} f^j + \tfrac{1}{2} e_i^p e_i^q \partial_{x^p} \partial_{x^q} f^j + \partial_{x^j} a^p f^p + \partial_{x^j} e_i^p e_i^q \partial_{x^q} f^p) \Delta t \\
& + (e_i^p \partial_{x^p} f^j + \partial_{x^j} e_i^p f^p) \Delta \eta_i \Big\} dx^j
\end{aligned} \tag{15}
$$

**Example 3.1.5.** When $\theta = f \frac{\partial}{\partial x^1} \wedge \cdots \wedge \frac{\partial}{\partial x^n}$,

$$
\begin{aligned}
T_{t*} \theta = \Big\{ & f + \Big((\partial_{x^p} a^p + \tfrac{1}{2} J_i) f + (-(a^p + e_i^p \partial_{x^q} e_i^q) + \partial_{x^q} e_i^p e_i^q) \partial_{x^p} f + \tfrac{1}{2} e_i^p e_i^q \partial_{x^p} \partial_{x^q} f\Big) \Delta t \\
& + (\partial_{x^p} e_i^p f - e_i^p \partial_{x^p} f) \Delta \eta_i \Big\} \frac{\partial}{\partial x^1} \wedge \cdots \wedge \frac{\partial}{\partial x^n}
\end{aligned} \tag{16}
$$

### 3.2 Derivation of the Stochastic PDE

Suppose $S$ is the full state variable of the dynamical system:

$$\frac{\partial S}{\partial t} = g(S). \tag{17}$$

Let $f$ be a component or a collection of components of $S$. We then associate $f$ to a differential form $\theta$ in the perturbation scheme, i.e. there is an invertible map $\mathcal{F}$ that maps the space of $f$ to the space of $\theta$, such that $\mathcal{F}(f) = \theta$. In the examples in this manuscript, the corresponding map $\mathcal{F}$ is obvious. When $f$ refers to a scalar quantity on the domain. We can choose to associate to $f$ a differential $n-$form $\theta = f dx^1 \wedge \cdots \wedge dx^n$ as in example 3.1.3, an $n-$vector $\theta = f \frac{\partial}{\partial x^1} \wedge \cdots \wedge \frac{\partial}{\partial x^n}$ as in example 3.1.5, or a differential $0-$form (function) $\theta = f$. When $f$ refers to a vector-valued function $f = (f^1, ..., f^n)$, we can associate to $f$ the differential $1-$form $f = f^1 dx^1 + \cdots + f^n dx^n$. It is not hard to see that $\mathcal{F}$ is obvious once the type of tensor field is chosen. Suppose the propagation equation for $f$ is

$$\mathrm{d}f = g^f(S) \mathrm{d}t. \tag{18}$$

This implies a propagation equation for $\theta$:

$$d\theta = g^\theta(S)dt. \tag{19}$$

The discrete-time perturbed forecast at each time step consists of the following two steps:

$$\begin{cases} \tilde{\theta}(t+\Delta t) = \theta(t) + g^\theta(S(t))\Delta t & (20) \\ \theta(t+\Delta t) = T_t^*\tilde{\theta}(t+\Delta t) & (21) \end{cases}$$

with $T_t^*\tilde{\theta}(t+\Delta t) = \tilde{\theta}(t+\Delta t) + \mathcal{M}(\tilde{\theta}(t+\Delta t))\Delta t + \mathcal{N}_i(\tilde{\theta}(t+\Delta t))\Delta\eta_i + o(\Delta t)$ for some differential forms $\mathcal{M}(\tilde{\theta})$ and $\mathcal{N}_i(\tilde{\theta})$.

As the physical PDE (20) is deterministic, $\|\tilde{\theta}(t+\Delta t) - \theta(t)\|$ scales in $O(\Delta t)$. Indeed, there is no noise term to induce a scaling in $O(\sqrt{\Delta t})$. Therefore, it can be assumed that there exists $C > 0$ so that $\|\mathcal{M}(\tilde{\theta}(t+\Delta t)) - \mathcal{M}(\theta(t))\| < C\Delta t$ and $\|\mathcal{N}_i(\tilde{\theta}(t+\Delta t)) - \mathcal{N}_i(\theta(t))\| < C\Delta t$, for $\Delta t$ small enough. Then

$$\begin{aligned} T_t^*\tilde{\theta}(t+\Delta t) &= \tilde{\theta}(t+\Delta t) + \Big(\mathcal{M}(\theta(t)) + \mathcal{O}(\Delta t)\Big)\Delta t + \Big(\mathcal{N}_i(\theta(t)) + \mathcal{O}(\Delta t)\Big)\Delta\eta_i + o(\Delta t) \\ &= \tilde{\theta}(t+\Delta t) + \mathcal{M}(\theta(t))\Delta t + \mathcal{N}_i(\theta(t))\Delta\eta_i + o(\Delta t) \end{aligned} \tag{22}$$

Therefore,

$$\theta(t+\Delta t) = \theta(t) + g^\theta(S(t))\Delta t + \mathcal{M}(\theta(t))\Delta t + \mathcal{N}_i(\theta(t))\Delta\eta_i + o(\Delta t). \tag{23}$$

This suggests the following stochastic propagation equation for $\theta$:

$$d\theta = g^\theta(S)dt + \mathcal{M}(\theta)dt + \mathcal{N}_i(\theta)d\eta_i. \tag{24}$$

Since there is a 1-1 correspondence between $\theta$ and $f$, Eq.(19) also suggests a stochastic propagation equation for $f$, which can be written as

$$df = g^f(S)dt + \mathcal{M}^f(f)dt + \mathcal{N}_i^f(f)d\eta_i. \tag{25}$$

We denote the additional terms in Eq.(25) by

$$d_s f := \mathcal{M}^f(f)dt + \mathcal{N}_i^f(f)d\eta_i. \tag{26}$$

Then Eq.(25) can be written as:

$$df = g^f(S)dt + d_s f. \tag{27}$$

**Remark 3** ($d_s f$ is not directly related to the original dynamics). $d_s f$ *is completely determined by $T_t^*\theta$, but is not directly related to the original dynamics Eq.(18). Therefore, once the expression of $T$ in Eq.(6) and the choice of $\theta$ is determined, the perturbation term $d_s f$ is prescribed. However, the choice of $\theta$ is up to the user, and may then be related to the original dynamics.*

**Remark 4.** *In particular, there is no noise in the the original dynamics Eq.(18) which could be correlated with the noise of the resulting stochastic scheme (21). That is why the Itō lemma directly applies in the Taylor development (B4) of $f$, and then in the equation (22), leading to (23) and the final SPDE. Indeed, unlike the Itō-Wentzell formula Kunita (1997) – a cornerstone of the LU scheme – there is no additional cross-correlation term between $T_t^*$ and $\tilde{\theta}(t + \Delta t)$. The final SPDE (24) makes clear the link between the solution $\theta$ and the Brownian motions $\eta_i$. But, at a given time step $t$, since (18) has no noise term, $\tilde{\theta}(t + \Delta t)$ is correlated with the $t' \mapsto \eta_i(t')$ for $t' < t$ only, and is independent of the new Brownian increment $\Delta \eta_i(t)$ generating $T_t$. Therefore, there is no cross-correlation term between $T_t^*$ and $\tilde{\theta}(t + \Delta t)$.*

**Remark 5.** *For a numerical implementation of our stochastic scheme, the time integration of the SPDE (25) may require a smaller time step $\Delta t$ than the time integration of the deterministic PDE (18) for two reasons. First, the available SDE time integration schemes are often less accurate than their deterministic counterparts. Secondly, the modified dynamics may involve additional CFL constraints, related to for instance noise-induced diffusion.*

**Example 3.2.1.** When $\theta = f$, example 3.1.1,

$$T_t^* \theta - \theta = \left( a^p \partial_{x^p} f + \tfrac{1}{2} e_i^p e_i^q \partial_{x^p} \partial_{x^q} f \right) \Delta t + e_i^p \partial_{x^p} f \Delta \eta_i \tag{28}$$

. This implies that

$$\mathsf{d}_s f = \left( a^p \partial_{x^p} f + \tfrac{1}{2} e_i^p e_i^q \partial_{x^p} \partial_{x^q} f \right) \mathsf{d}t + e_i^p \partial_{x^p} f \mathsf{d}\eta_i. \tag{29}$$

To physically interpret this equation, we rewrite:

$$\frac{\mathsf{d}_s f}{\mathsf{d}t} + V^p \partial_{x^p} f = \partial_{x^p} \left( (\tfrac{1}{2} e_i^p e_i^q) \partial_{x^q} f \right) \tag{30}$$

where

$$V^p = -a^p + \tfrac{1}{2} \partial_{x^q} (e_i^p e_i^q) - e_i^p \frac{\mathsf{d}\eta_i}{\mathsf{d}t}. \tag{31}$$

Terms of advection and diffusion are recognized. The matrix $\tfrac{1}{2} e_i e_i^T$ is symmetric non-negative and represents a diffusion matrix. The $p$-th component of the advecting velocity $V^p$ is composed of the drift $-a^p$, a correction $\tfrac{1}{2} \partial_{x^q} (e_i^p e_i^q)$, and a stochastic advecting velocity $-e_i^p \frac{\mathsf{d}\eta_i}{\mathsf{d}t}$.

If the original deterministic PDE (18) is an advection diffusion equation, with advecting velocity $u$ and diffusion coefficient coefficient $D$, the final SPDE to simulate (Eq. (25)) is now a stochastic advection-diffusion equation, with advecting velocity $u + V$ and diffusion matrix $DI_d + \tfrac{1}{2} e_i e_i^T$:

$$\frac{\mathsf{d}f}{\mathsf{d}t} + (u^p + V^p) \partial_{x^p} f = \partial_{x^p} \left( (D\delta_{pq} + \tfrac{1}{2} e_i^p e_i^q) \partial_{x^q} f \right) \tag{32}$$

This type of SPDE appears in the LU framework, detailed in section 4.2.1.

**Example 3.2.2.** When $\theta = f\,dx^1 \wedge \cdots \wedge dx^n$, example 3.1.3,

$$T_t^*\theta - \theta = \left\{ \left( (\partial_{x^p} a^p + \tfrac{1}{2} J_i) f + (a^p + e_i^p \partial_{x^q} e_i^q) \partial_{x^p} f + \tfrac{1}{2} e_i^p e_i^q \partial_{x^p} \partial_{x^q} f \right) \Delta t \right.$$
$$\left. + (\partial_{x^p} e_i^p f + e_i^p \partial_{x^p} f) \Delta \eta_i \right\} dx^1 \wedge \cdots \wedge dx^n \tag{33}$$

This implies that

$$\mathrm{d}_s f = \left( (\partial_{x^p} a^p + \tfrac{1}{2} J_i) f + (a^p + e_i^p \partial_{x^q} e_i^q) \partial_{x^p} f + \tfrac{1}{2} e_i^p e_i^q \partial_{x^p} \partial_{x^q} f \right) \mathrm{d}t$$
$$+ (\partial_{x^p} e_i^p f + e_i^p \partial_{x^p} f) \mathrm{d}\eta_i \tag{34}$$

Rewritten, it leads to:

$$\frac{\mathrm{d}_s f}{\mathrm{d}t} + \partial_{x^p} \left( \tilde{V}^p f \right) = \partial_{x^p} \left( (\tfrac{1}{2} e_i^p e_i^q) \partial_{x^q} f \right) \tag{35}$$

where

$$\tilde{V}^p = V^p - (e_i^p \partial_{x^q} e_i^q) = -a^p + \tfrac{1}{2}(\partial_{x^q} e_i^p e_i^q - e_i^p \partial_{x^q} e_i^q) - e_i^p \frac{\mathrm{d}\eta_i}{\mathrm{d}t} \tag{36}$$

Again an advection-diffusion equation is recognized, but of different nature. Indeed, as expected for an n-form, the PDE is similar to a density conservation equation. Moreover, the advecting drift is slightly different to take into account the cross-correlations between $f(T_t(x))$ and $T_t^*(dx^1 \wedge \cdots \wedge dx^n)$.

Recall, in fluid dynamics, the Reynolds transport theorem provides an integral conservation equation for the transport of any conserved quantity within a fluid, connected to its corresponding differential equation. The Reynolds transport theorem is central to the LU setting. The present example thus already outlines a closed link between the proposed perturbation approach and the LU formulation. Accordingly, the SPDE (35) naturally appears in the LU framework, as detailed in section 4.2.2.

**Example 3.2.3.** When $\theta = f^j\,dx^j$, example 3.1.4,

$$T_t^*\theta - \theta = \left\{ (a^p \partial_{x^p} f^j + \tfrac{1}{2} e_i^p e_i^q \partial_{x^p} \partial_{x^q} f^j + \partial_{x^j} a^p f^p + \partial_{x^j} e_i^p e_i^q \partial_{x^q} f^p) \Delta t \right.$$
$$\left. + (e_i^p \partial_{x^p} f^j + \partial_{x^j} e_i^p f^p) \Delta \eta_i \right\} dx^j \tag{37}$$

For each $j$, the coefficients of $dx^j$ in $T_t^*\theta - \theta$ and those in $\theta$ can be compared, to lead to

$$\mathrm{d}_s f^j = (a^p \partial_{x^p} f^j + \tfrac{1}{2} e_i^p e_i^q \partial_{x^p} \partial_{x^q} f^j + \partial_{x^j} a^p f^p + \partial_{x^j} e_i^p e_i^q \partial_{x^q} f^p) \mathrm{d}t$$
$$+ (e_i^p \partial_{x^p} f^j + \partial_{x^j} e_i^p f^p) \mathrm{d}\eta_i \tag{38}$$

Regrouping the terms for physical interpretation, it writes:

$$\frac{\mathrm{d}_s f^j}{\mathrm{d}t} + V^p \partial_{x^p} f^j + \partial_{x^j} \left( -a^p - e_i^p \frac{\mathrm{d}\eta_i}{\mathrm{d}t} \right) f^p - \partial_{x^j} e_i^p e_i^q \partial_{x^q} f^p = \partial_{x^p} \left( (\tfrac{1}{2} e_i^p e_i^q) \partial_{x^q} f^j \right) \tag{39}$$

Two additional terms complete the advection-diffusion term. The first one, $\partial_{x^j} \left( -a^p - e_i^p \frac{\mathrm{d}\eta_i}{\mathrm{d}t} \right) f^p$, is reminiscent of the additional terms appearing in SALT momentum equations Holm (2015); Resseguier et al. (2020). The second term, $-\partial_{x^j} e_i^p e_i^q \partial_{x^q} f^p$, comes from the cross-correlation in Itô notation.

**Example 3.2.4.** When $\theta = f \frac{\partial}{\partial x^1} \wedge \cdots \wedge \frac{\partial}{\partial x^n}$, example 3.1.5,

$$T_{t*}\theta - \theta = \Big\{ \Big( (\partial_{x^p} a^p + \tfrac{1}{2} J_i)f + (-(a^p + e_i^p \partial_{x^q} e_i^q) + \partial_{x^q} e_i^p e_i^q)\partial_{x^p} f + \tfrac{1}{2} e_i^p e_i^q \partial_{x^p} \partial_{x^q} f \Big)\Delta t$$
$$+ (\partial_{x^p} e_i^p f - e_i^p \partial_{x^p} f)\Delta \eta_i \Big\} \frac{\partial}{\partial x^1} \wedge \cdots \wedge \frac{\partial}{\partial x^n} \tag{40}$$

This implies

$$\mathtt{d}_s f = \Big( (\partial_{x^p} a^p + \tfrac{1}{2} J_i)f + (-(a^p + e_i^p \partial_{x^q} e_i^q) + \partial_{x^q} e_i^p e_i^q)\partial_{x^p} f + \tfrac{1}{2} e_i^p e_i^q \partial_{x^p} \partial_{x^q} f \Big)\mathtt{d}t$$
$$+ (\partial_{x^p} e_i^p f - e_i^p \partial_{x^p} f)\mathtt{d}\eta_i \tag{41}$$

It can then be verified that:

$$\frac{\mathtt{d}_s f}{\mathtt{d}t} + \partial_{x^p} \tilde{V}^p f - \tilde{\tilde{V}}^p \partial_{x^p} f = \partial_{x^p}\left( (\tfrac{1}{2} e_i^p e_i^q)\partial_{x^q} f \right) \tag{42}$$

where

$$\tilde{\tilde{V}}^p = \tilde{V}^p - (e_i^p \partial_{x^q} e_i^q) = V^p - 2(e_i^p \partial_{x^q} e_i^q) \tag{43}$$

We recognize a diffusion term, $\partial_{x^p}\left( (\tfrac{1}{2} e_i^p e_i^q)\partial_{x^q} f \right)$, a velocity divergence term, $\partial_{x^p} \tilde{V}^p f$, and the advection term, $-\tilde{\tilde{V}}^p \partial_{x^p} f$. The divergence term is comparable to one appearing in the density equation. However, the velocity fields appearing in the divergent and advecting terms do not coincide. Indeed, they are even opposite for divergence-free noise ($\partial_{x^q} e_i^q = 0$). This type of equation may appear uncommon but will be shown useful when applied to randomized thermal shallow water equations.

### 3.3 Conservation laws related to $\mathtt{d}_s f$

A major advantage of the proposed perturbation scheme is to possibly prescribe $\theta$ to ensure that certain quantities are conserved. Define the discrete time version of $\mathtt{d}_s f$ as:

$$\Delta_s f = \mathcal{M}^f(f)\Delta t + \mathcal{N}_i^f(f)\Delta \eta_i. \tag{44}$$

In general, conservation laws can be derived from the following two identities about the pull-back operator:

$$(T_t^* \theta_1) \wedge (T_t^* \theta_2) = T_t^*(\theta_1 \wedge \theta_2) \tag{45}$$
$$dT_t^* \theta = T_t d\theta, \tag{46}$$

where $d$ refers to the differential operator acting on differential forms. Hereafter, we present how to derive the conservation laws for two particular examples.

**Example 3.3.1.** Suppose $\theta_1 = f dx^1 \wedge \cdots \wedge dx^n$ and define

$$\hat{\theta}_1 = T_t^* \theta_1 \tag{47}$$
$$\hat{f} = f + \Delta_s f. \tag{48}$$

Then $\hat{\theta}_1 = \hat{f}dx^1 \wedge \cdots \wedge dx^n$. Therefore

$$\int_\Omega \hat{f}dx^1 \ldots dx^n = \int_\Omega \hat{\theta}_1 = \int_\Omega T_t^* \theta_1 = \int_{T_t(\Omega)} \theta_1 = \int_\Omega \theta_1$$

$$= \int_\Omega f dx^1 \ldots dx^n. \tag{49}$$

Eq.(49) implies that the total integral of $f$ is not changed by the perturbation scheme. Next suppose that $\theta_2 = g$ is a function. Similarly we define

$$\hat{\theta}_2 = T_t^* \theta_2 \tag{50}$$

$$\hat{g} = g + \Delta_s g. \tag{51}$$

Applying Eq.(45),

$$\int_\Omega \hat{f}\hat{g}dx^1 \ldots dx^n = \int_\Omega \hat{\theta}_1 \wedge \hat{\theta}_2 = \int_\Omega T_t^*(\theta_1 \wedge \theta_2) = \int_{T_t(\Omega)} \theta_1 \wedge \theta_2$$

$$= \int_\Omega \theta_1 \wedge \theta_2 = \int_\Omega fg dx^1 \ldots .dx^n \tag{52}$$

The total integral of $fg$ is thus also conserved by the perturbation scheme. Similarly for any integer $m \geq 0$, $fg^m$ is conserved by the perturbation scheme.

**Example 3.3.2.** Suppose $n = 2$ and $\theta = udx + vdy$, where $\mathbf{u} = (u, v)$ is the velocity field. The vorticity $\omega = \partial_x v - \partial_y u$ corresponds to the differential 2-form $d\theta$:

$$d\theta = \omega dx^1 \wedge dx^2. \tag{53}$$

Define $\hat{\theta} := T_t^* \theta = \hat{u}dx^1 + \hat{v}dx^2$ and $\hat{\omega} = \partial_x \hat{v} - \partial_y \hat{u}$. Then $d\hat{\theta} = \hat{\omega}dx^1 \wedge dx^2$, and

$$\int_\Omega \hat{\omega}dx^1 dx^2 = \int_\Omega d\hat{\theta} = \int_\omega dT_t^* \theta = \int_\Omega T_t^* d\theta = \int_{T_t(\Omega)} d\theta$$

$$= \int_\Omega \omega dx^1 dx^2. \tag{54}$$

Therefore the vorticity is conserved by the perturbation scheme.

**Example 3.3.3.** Suppose $n = 3$ and $\theta = udx + vdy + wdz$, where $\mathbf{u} = (u, v, w)$ is the velocity field. The vorticity $\omega = (\partial_y w - \partial_z v, \partial_z u - \partial_x w, \partial_x v - \partial_y u)$ corresponds to the differential 2-form $d\theta$:

$$d\theta = (\partial_y w - \partial_z v)dy \wedge dz + (\partial_x v - \partial_y u)dz \wedge dx + (\partial_x v - \partial_y u)dx \wedge dy. \tag{55}$$

The helicity $\Theta = u(\partial_y w - \partial_z v) + v(\partial_x v - \partial_y u) + w(\partial_x v - \partial_y u)$ corresponds to the differential 3-form:

$$d\theta \wedge \theta = \Big( u(\partial_y w - \partial_z v) + v(\partial_x v - \partial_y u) + w(\partial_x v - \partial_y u) \Big) dx \wedge dy \wedge dz. \tag{56}$$

Similarly, we define $\hat\Theta$ by $d\hat\theta \wedge \hat\theta = \hat\Theta dx \wedge dy \wedge dz$. Then

$$\int_\Omega \hat\Theta dx dy dz = \int_\Omega d\hat\theta \wedge \hat\theta = \int_\Omega (dT_t^*\theta) \wedge (T_t^*\theta)$$

$$= \int_\Omega (T_t^* d\theta) \wedge (T_t^*\theta) = \int_\Omega T_t^*(d\theta \wedge \theta) = \int_{T_t(\Omega)} d\theta \wedge \theta = \int_\Omega \Theta dx dy dz. \tag{57}$$

Hence, in this case, the total amount of helicity is conserved.

**Example 3.3.4.** Suppose that $\theta_1 = f dx^1 \wedge \cdots \wedge dx^n$ and that $\theta_2 = g \frac{\partial}{\partial x^1} \wedge \cdots \wedge \frac{\partial}{\partial x^n}$. There exists a pairing $\langle , \rangle$ for the differential $n-$forms and the contravariant $n-$vectors, i.e. $\langle \theta_1, \theta_2 \rangle = fg$ is a function on $\Omega$. Define

$$\hat\theta_1 = T_t^*\theta_1 = \hat f dx^1 \wedge \cdots \wedge dx^n \tag{58}$$

$$\hat\theta_2 = (T_t^{-1})_*\theta_2 = \hat g \frac{\partial}{\partial x^1} \wedge \cdots \wedge \frac{\partial}{\partial x^n} \tag{59}$$

Then we have

$$\hat f \hat g(T_t^{-1}(x)) = \langle \hat\theta_1, \hat\theta_2 \rangle \big|_{T_t^{-1}(x)} = \langle \theta_1, \theta_2 \rangle \big|_x = fg(x), \tag{60}$$

and that

$$\int_\Omega \hat f^2 \hat g \, dx^1 \ldots dx^n = \int_\Omega \langle \hat\theta_1, \hat\theta_2 \rangle \theta_1 = \int_\Omega \langle \theta_1, \theta_2 \rangle \theta_1 = \int_\Omega f^2 g \, dx^1 \ldots dx^n \tag{61}$$

**Remark 6** (The conservation law of the perturbation scheme is independent of the conservation law of the original dynamical
system)**. *The derivation of Eqs.(49) (52), (54), (57), and (61) is based on the generic properties of the pull-back and push-
forward operator of tensor fields. Since the choice of $\theta$ is not directly determined by the dynamical system, the conservation
law of the perturbation scheme is independent of the original dynamical system. Recall that the perturbed forecast consists of
two steps: Eq.(20) and (21). The conservation law of the perturbation scheme implies that certain quantities are conserved
in the second step. On the other hand, the original dynamical system Eq.(20) might enjoy some other conservation law. If a
350 quantity is conserved by both the original dynamical system and the perturbation scheme, then this quantity must be conserved
by the final stochastic PDE. If a quantity is conserved by only one of Eqs.(20) and (21), then it can not be concluded that this
quantity is conserved by the final SPDE.*

# 4 Comparison with other perturbation schemes

In this section, we demonstrate that both the stochastic advection by Lie transport (SALT) equation Holm (2015) and the loca-
355 tion uncertainty (LU) equation Mémin (2014); Resseguier et al. (2017a, 2020) can be recovered using the proposed perturbation
scheme and properly choosing $\theta$ and the parameters $a, e_i$.

Note that the original LU paper (Mémin, 2014) assumed strong smoothness properties (finite variations in time) of the stochastic Navier-Stokes equations solution, to eventually remove the noises terms of this original Navier-Stokes equations under location uncertainty. Since Resseguier et al. (2017a, b), this assumption was removed, in order to keep the important noise terms. Accordingly, the original deterministic LU Navier-Stokes equations from Mémin (2014) have been referred to as pseudo-stochastic Navier-Stokes equations (Resseguier et al., 2021). Being deterministic, this pseudo-stochastic equations cannot be recovered by our stochastic scheme, whereas we can recover the stochastic LU Navier-Stokes equations, originated from Resseguier et al. (2017a).

## 4.1 Comparison with SALT equation

The original SALT equation Holm (2015) is derived based on a stochastically constrained variational principle $\delta S = 0$, for which

$$
\begin{cases}
S(u,q) = \int \ell(u,q)\mathrm{d}t \\
\mathrm{d}q + \pounds_{\mathrm{d}x_t} q = 0.
\end{cases}
\tag{62}
$$

where $\ell(u,q)$ is the Lagrangian of the system, $\pounds$ is the Lie derivative, and $x_t(x)$ is defined by (using our notation)

$$
x_t(x) = x_0(x) + \int\limits_0^t u(x,s)\mathrm{d}s - \int\limits_0^t e_i(x) \circ \mathrm{d}\eta_i(s),
\tag{63}
$$

in which $u$ is the velocity vector field, and the $\circ$ means that the integral is defined in the Stratonovich sense, instead of in the Itô sense. Hence, $\mathrm{d}x_t = u(x,t)\mathrm{d}t - e_i \circ \mathrm{d}\eta_i$ refers to an infinitesimal stochastic tangent field on the domain. Broadly speaking, we can express $\mathrm{d}x_t = T_t(x) - x + u\mathrm{d}t$. Note the difference between Itô's notation and Stratonovich's notation, i.e. $e_i \circ \mathrm{d}\eta_i \neq e_i \mathrm{d}\eta_i$. Our expression of $T_t$ essentially follows Itô's notation, and $T_t(x) \neq x - e_i\Delta\eta_i$ in this subsection. Instead, it becomes $T_t(x) = x + \frac{1}{2}e_i^p \partial_{x_p} e_i \Delta t - e_i \Delta\eta_i$.

In the second equation of Eq.(62), $q$ is assumed to be a quantity advected by the flow. $q$ can correspond to any differential form that is not uniquely determined by the velocity (since the SALT equation for the velocity is usually determined by the first equation of Eq.(62)). In Holm (2015), the Lie derivative $\pounds_{\mathrm{d}x_t} q$ is calculated using Cartan's formula:

$$
\pounds_{\mathrm{d}x_t} q = d(i_{\mathrm{d}x_t} q) + i_{\mathrm{d}x_t} dq.
\tag{64}
$$

Essentially, the Lie derivative $\pounds_{\mathrm{d}x_t} q$ corresponds to $T_t^* q - q + f^q(S)\mathrm{d}t$, if we assume that the deterministic forecast of $q$ is simply the advection of $q$ by $u$. More generally, $\pounds_{\mathrm{d}x_t - u\mathrm{d}t} q = T_t^* q - q$. Therefore, the SALT equation for $q$ is the same as our equation for $q$. We remark that the Cartan's formula can not be directly applied to calculate the Lie derivative if the expression of $\mathrm{d}x_t$ is in Itô's notation.

The SALT equation regarding the velocity $u$ comes from the first equation of Eq.(62). For most cases, the velocity $u$ is associated with the momentum, a differential $1-$form $\mathrm{m} = u^j dx^j = u^1 dx^1 + ... + u^n dx^n$. In the examples discussed in Holm (2015), it is observed that, when the Lagrangian includes the kinetic energy, the stochastic noise contribute a term $\pounds_{\mathrm{d}x_t}\theta$, where

$\theta$ is a differential $1-$form related to the momentum $1-$form. For instance, $\theta = m$ in the example of "Stratonovich stochastic Euler-Poincaré flow" in Holm (2015), and $\theta = m + R^j dx^j$ in the example of "Stochastic Euler-Boussinesq equations of a rotating stratified incompressible fluid" in Holm (2015). Already pointed out, the operator $\mathcal{L}_{dx_t}$ is closely related to $T_t^*$, and the momentum equation in SALT can be derived using our proposed scheme by properly choosing $\theta$.

Holm (2015) requires that $q$ to be a differential form since Cartan's formula is only useful for differential forms $q$. This restriction can be relaxed by employing the original definition of Lie derivative with respect to a deterministic/stochastic flow of diffeomorphism discussed in Leon (2021), so that $\mathcal{L}_{dx_t} q$ can be generalized to the case where $q$ is a mixed tensor field. This corresponds to our Eq.(7).

Compared with Holm (2015); Leon (2021), the proposed perturbation approach seems more flexible and does not have to rely on the Lagrangian mechanics. In particular, the velocity field can be associated to other tensor fields than the momentum 1-form. The perturbation, not directly related to the physics, can then be applied to any PDE. Moreover, our approach provides a new interpretation of $\mathcal{L}_{dx_t - u dt}$ in terms of the optimal transportation associated with the infinitesimal forecast error at each time step. This interpretation certainly suggests practical numerical methods to infer $a, e_i$. Given a long sequence of reanalysis data or simulated high-resolution data, the one-step forecast can be evaluated using the low resolution model, with the high resolution state at each time step being the initial condition. $T_t$ is then estimated at each time step by comparing the low resolution forecast and the high resolution forecast. Finally, $a$ and $e_i$ could be learnt from these samples of $T_t$.

## 4.2 Comparison with the LU equation

Mentioned above, the Reynolds transport theorem is central to the LU setting, and we already outlined a closed link between the proposed perturbation approach and the LU formulation. This link – related to differential $n-$forms – will be precised later in this subsection. But, before this, we focus on another key ingredient of LU: the stochastic material derivative of functions (differntial $0-$forms).

### 4.2.1 0-forms in the LU framework

Dropping the forcing terms, the LU equation for compressible and incompressible flow writes [Resseguier et al. (2017a)].

$$\partial_t f + \boldsymbol{w}^\star \cdot \nabla f = \nabla \cdot (\tfrac{1}{2} \boldsymbol{a} \nabla f) - \boldsymbol{\sigma} \dot{\boldsymbol{B}} \cdot \nabla f \tag{65}$$

$$\boldsymbol{w}^\star = \boldsymbol{w} - \tfrac{1}{2} (\nabla \cdot \boldsymbol{a})^\top + \boldsymbol{\sigma} (\nabla \cdot \boldsymbol{\sigma})^\top, \tag{66}$$

where $f$ can be any quantity that is assumed to be transported by the flow, i.e. $Df/Dt = 0$ where $D/Dt$ is the Itō material derivative. For instance, $f$ could be the velocity (dropping forces in the SPDE), the temperature, or the buoyancy. Compared to SALT notations, $-e_i d\eta_i$ is denoted $\boldsymbol{\sigma} d\boldsymbol{B} = \boldsymbol{\sigma}_{\bullet i} dB_i$. We refer to (Resseguier et al., 2020, Appendix A) for the complete table of SALT-LU notations correspondences. Derived in (Resseguier, 2017, Appendix 10.1) and (Resseguier et al., 2021, 6.1.3), we

can rewrite it as

$$\partial_t f + \boldsymbol{w}_S \cdot \nabla f = \tfrac{1}{2}(\boldsymbol{\sigma}_{\bullet i} \cdot \nabla)(\boldsymbol{\sigma}_{\bullet i} \cdot \nabla f) - (\boldsymbol{\sigma} \dot{\boldsymbol{B}}) \cdot \nabla f, \tag{67}$$

$$= -(\boldsymbol{\sigma} \circ \dot{\boldsymbol{B}}) \cdot \nabla f, \tag{68}$$

$$\boldsymbol{w}_S = \boldsymbol{w} + \boldsymbol{w}_S^c \tag{69}$$

$$\boldsymbol{w}_S^c = -\tfrac{1}{2}(\nabla \cdot \boldsymbol{a})^\top + \tfrac{1}{2}\boldsymbol{\sigma}(\nabla \cdot \boldsymbol{\sigma})^\top, \tag{70}$$

$$= -\tfrac{1}{2}(\boldsymbol{\sigma}_{\bullet i} \cdot \nabla)\boldsymbol{\sigma}_{\bullet i},, \tag{71}$$

where $\boldsymbol{\sigma} \circ \dot{\boldsymbol{B}}$ is the Stratonovich noise of the SPDE, $\boldsymbol{w}$ and $\boldsymbol{w}_S$ (denoted $u$ in the SALT framework) are respectively the Itō drift and the Stratonovich drift of the fluid flow. Separating the terms of the SPDE related to the deterministic dynamics from the term associated to the stochastic scheme, it comes

$$\mathrm{d}^{\mathrm{LU}} f = g^f(S)\mathrm{d}t + \mathrm{d}_s^{\mathrm{LU}} f, \tag{72}$$

where

$$g^f(S) = -\boldsymbol{w} \cdot \nabla f \tag{73}$$

$$\mathrm{d}_s^{\mathrm{LU}} f = -\boldsymbol{w}_S^c \cdot \nabla f \mathrm{d}t + \tfrac{1}{2}(\boldsymbol{\sigma}_{\bullet i} \cdot \nabla)(\boldsymbol{\sigma}_{\bullet i} \cdot \nabla f)\mathrm{d}t - (\boldsymbol{\sigma} \mathrm{d}\boldsymbol{B}) \cdot \nabla f \tag{74}$$

Terms in Eqs.(65) and (66) translate to our notation in the following way:

$$-\boldsymbol{w}_S^c \cdot \nabla f \mathrm{d}t = \tfrac{1}{2}e_i^q \partial_{x_q} e_i^p \partial_{x_p} f$$

$$\tfrac{1}{2}(\boldsymbol{\sigma}_{\bullet i} \cdot \nabla)(\boldsymbol{\sigma}_{\bullet i} \cdot \nabla f) = \tfrac{1}{2}e_i^p \partial_{x_p}(e_i^q \partial_{x_q} f)$$

$$= \tfrac{1}{2}(e_i^p \partial_{x_p} e_i^q \partial_{x_q} f + e_i^p e_i^q \partial_{x_p} \partial_{x_q} f)$$

$$-\boldsymbol{\sigma} \mathrm{d}\boldsymbol{B} \cdot \nabla f = e_i^p \partial_{x_p} f \mathrm{d}\eta_i$$

Hence

$$\mathrm{d}_s^{\mathrm{LU}} f = (e_i^q \partial_{x_q} e_i^p \partial_{x_p} f + \tfrac{1}{2}e_i^p e_i^q \partial_{x_p} \partial_{x_q} f)\mathrm{d}t + e_i^p \partial_{x_p} f \mathrm{d}\eta_i \tag{75}$$

Recall that Eq.(29) can be obtained by our perturbation scheme while $f$ is associated to a differential $0-$form. Direct calculation yields that Eq.(75) coincides with Eq.(29) when

$$T_t(x) = x + e_i^q \partial_{x_q} e_i \Delta t + e_i \Delta \eta_i = x - \boldsymbol{w}_S^c \Delta t + (-\boldsymbol{w}_S^c \Delta t - \boldsymbol{\sigma} \Delta \boldsymbol{B}). \tag{76}$$

The LU equation can thus be derived by choosing $\theta = f$ and $T_t$ by Eq.(76). At the first glance, it seems not straightforward to make such a choice. Nevertheless, it can be recognized that the term $(-\boldsymbol{w}_S^c \Delta t - \boldsymbol{\sigma} \Delta \boldsymbol{B}) = (\tfrac{1}{2}e_i^q \partial_{x_q} e_i \Delta t + e_i \Delta \eta_i)$ is the Itō noise plus its Itō-to-Stratonovich correction. Hence, it corresponds to the Stratonovich noise $e_i \circ \mathrm{d}\eta_i$ of the flow associated to $T_t$. The additional drift $-\boldsymbol{w}_S^c \Delta t$ is different in nature. It is related to the advection correction $\boldsymbol{w}_S^c \cdot \nabla f$ in the LU setting.

Indeed, in the LU framework, the Itō drift, $\boldsymbol{w}$, is seen as the resolved large-scale velocity. That is why, in this framework, the deterministic dynamics (74) involves the Itō drift, $\boldsymbol{w}$. This is also the reason why, under the LU derivation, the advected velocity is assumed to be given by the Itō drift, $\boldsymbol{w}$. It differs from the Stratonovich drift $\boldsymbol{w}_S = \boldsymbol{w} + \boldsymbol{w}_S^c$, used as an advection velocity in the SALT approach or in Mikulevicius and Rozovskii (2004) (where the Stratonovich drift is denoted $u$). Interested readers are referred to (Resseguier et al., 2020, Appendix A) for a discussion on these assumptions. Note however that in all these approaches, the advecting velocity is always the Stratonovich drift. This can be seen e.g., in the Stratonovich form of LU equations (68).

To also understand (76), the inverse flow can be considered. According to appendix A,

$$T_t^{-1}(x) = x - e_i \Delta \eta_i = x + \boldsymbol{\sigma} \Delta \boldsymbol{B}. \tag{77}$$

Considering $T_t$ to represent how much the model forecast differs from the true forecast at every time step, $T_t^{-1}$ can be understood to represent how much the true forecast differs from the model forecast at each time step. Therefore, the LU equation can be derived using the proposed perturbation scheme, choosing $\theta = f$ and assuming that the true forecast differs from the model forecast by a displacement prescribed by Eq.(77).

### 4.2.2   n-forms in the LU framework

The LU physical justification relies on a stochastic interpretation of fundamental conservation laws, typically conservation of extensive properties (i.e. integrals of functions over a spatial volume) like momentum, mass, matter and energy Resseguier et al. (2017a). These extensive properties can be expressed by integrals of differential $n-$forms. For instance, the mass and the momentum are integrals of the differential $n-$forms $\rho dx^1 \wedge \cdots \wedge dx^n$ and $\rho \boldsymbol{w} dx^1 \wedge \cdots \wedge dx^n$, respectively. In the LU framework, a stochastic version of the Reynolds transport theorem (Resseguier et al., 2017a, Eq. (28)) is used to deal with these differential $n-$forms $\theta = f dx^1 \wedge \cdots \wedge dx^n$. Assuming an integral conservation $\frac{d}{dt} \int_{V(t)} f = 0$ on a spatial domain $V(t)$ transported by the flow, that theorem leads to the following SPDE:

$$\frac{Df}{Dt} + \nabla \cdot (\boldsymbol{w}^\star + \boldsymbol{\sigma} \dot{\boldsymbol{B}}) f = \frac{d}{dt} \left\langle \int_0^t D_t f, \int_0^t \nabla \cdot \boldsymbol{\sigma} \dot{\boldsymbol{B}} \right\rangle = (\nabla \cdot \boldsymbol{\sigma}_{\bullet i})(\nabla \cdot \boldsymbol{\sigma}_{\bullet i})^T f \tag{78}$$

where $D/Dt$ denotes the Itō material derivative. Here again, forcing terms are dropped for the sake of readability. This SPDE can be rewritten using the expression of that material derivative (Eq. (9) and (10) of Resseguier et al. (2017a)):

$$\partial_t f + \nabla \cdot (\boldsymbol{w}_S f) = \tfrac{1}{2} \nabla \cdot (\boldsymbol{a} \nabla f) + \tfrac{1}{2} \nabla \cdot (\boldsymbol{\sigma}_{\bullet i}(\nabla \cdot \boldsymbol{\sigma}_{\bullet i})^T f) - \nabla \cdot (\boldsymbol{\sigma} \dot{\boldsymbol{B}} f) \tag{79}$$

$$= \tfrac{1}{2} \nabla \cdot (\boldsymbol{\sigma}_{\bullet i}(\nabla \cdot (\boldsymbol{\sigma}_{\bullet i} f))^T) - \nabla \cdot (\boldsymbol{\sigma} \dot{\boldsymbol{B}} f) \tag{80}$$

$$= - \nabla \cdot (\boldsymbol{\sigma} \circ \dot{\boldsymbol{B}} f) \tag{81}$$

The original deterministic equation and stochastic perturbation correspond to

$$g^f(S) = -\nabla \cdot (\boldsymbol{w} f) \tag{82}$$

$$d_s^{LU} f = (-\nabla \cdot (\boldsymbol{w}_S^c f) + \tfrac{1}{2}\nabla \cdot (\boldsymbol{a}\nabla f) + \tfrac{1}{2}\nabla \cdot (\boldsymbol{\sigma}_{\bullet i}(\nabla \cdot \boldsymbol{\sigma}_{\bullet i})^T f)) dt - \nabla \cdot (\boldsymbol{\sigma} d\boldsymbol{B} f) \tag{83}$$

$$= \nabla \cdot (((\tfrac{1}{2}\nabla \cdot \boldsymbol{a})^T dt - \boldsymbol{\sigma} d\boldsymbol{B}) f) + \nabla \cdot (\tfrac{1}{2}\boldsymbol{a}\nabla f) dt \tag{84}$$

Identifying $\boldsymbol{a} = \boldsymbol{\sigma}_{\bullet i}\boldsymbol{\sigma}_{\bullet i}^T = e_i e_i^T$ and $\boldsymbol{\sigma}\dot{\boldsymbol{B}} = -e_i d\eta_i$, Eq. (35) corresponds to example 3.2.2 about $n-$forms, with

$$\tilde{V} = -a^p + \tfrac{1}{2}(\partial_{x^q} e_i^p e_i^q - e_i^p \partial_{x^q} e_i^q) - e_i^p \frac{d\eta_i}{dt} = -(\tfrac{1}{2}\nabla \cdot \boldsymbol{a})^T + \boldsymbol{\sigma}\dot{\boldsymbol{B}} \tag{85}$$

i.e.

$$a^p = \partial_{x^q}(e_i^p e_i^q) - (e_i^p \partial_{x^q} e_i^q) = e_i^q \partial_{x^q} e_i^p. \tag{86}$$

Again the remapping is obtained

$$T_t(x) = x + e_i^q \partial_{x_q} e_i \Delta t + e_i \Delta \eta_i = x - \boldsymbol{w}_S^c \Delta t + (-\boldsymbol{w}_S^c \Delta t - \boldsymbol{\sigma}\Delta\boldsymbol{B}), \tag{87}$$

previously derived for differential $0-$form in LU framework (Eq. (76)). Therefore, the proposed approach also generalizes the LU framework for $n-$ forms, and its capacity – given by the Reynolds transport theorem – to deal with extensive properties.

**Remark 7.** *For incompressible flows, LU equation further imposes that*

$$\begin{cases} \nabla \cdot \boldsymbol{\sigma} = 0 \\ \nabla \cdot \nabla \cdot \boldsymbol{a} = 0 \end{cases} \tag{88}$$

*Translating it into our notation, it reads as*

$$\begin{cases} \partial_{x_p} e_i^p = 0 \, for \, each \, i \\ \partial_{x_p}\partial_{x_q}(e_i^p e_i^q) = 0 \end{cases}$$

*Applying the result in example 3.1.2, straightforward calculation gives Eq.(88) to be equivalent to that $T_t^*\theta = \theta$ for $\theta = dx^1 \wedge \cdots \wedge dx^n$. Such a result was expected since constraints Eq. (88) are obtained from the LU density conservation.*

## 5 A stochastic version of thermal shallow water equation

In this section, the proposed approach is applied to derive a stochastic version of thermal shallow water equation. Another stochastic version of thermal shallow water equation can be found in Holm and Luesink (2021). The thermal shallow water equation is derived in Warneford and Dellar (2013):

$$\frac{\partial h}{\partial t} + \nabla \cdot (h\bar{u}) = 0, \tag{89}$$

$$\frac{\partial \Theta}{\partial t} + (\bar{u} \cdot \nabla)\Theta = -\kappa(h\Theta - h_0\Theta_0), \tag{90}$$

$$\frac{\partial \bar{u}}{\partial t} + (\bar{u} \cdot \nabla)\bar{u} + f\hat{z} \times \bar{u} = -\nabla(h\Theta) + \frac{1}{2}h\nabla\Theta \tag{91}$$

This model can be used to describe a two-layer system under equivalent barotropic approximation. The upper layer is active but with a spatio-temporal varying density $\rho(x,t)$, while the lower layer is quiescent with a fixed constant density $\rho_0$. The state variable $h$ represents the height of the active layer, and $\Theta = g(\rho_0 - \rho)/\rho_0$ is the density contrast. $\bar{u}$ is the averaged horizontal velocity of the active layer at each column. Note that $\rho < \rho_0$ (hence $\Theta > 0$) in the scenario of equivalent barotropic approximation Warneford and Dellar (2013).

Stated in Warneford and Dellar (2013), the following physical quantities are conserved up to the forcing:

Total energy: $E = \displaystyle\int_\Omega \frac{1}{2}(h|\bar{u}|^2 + h^2\Theta)d^2x$ (92)

Total mass: $\mathcal{M} = \displaystyle\int_\Omega h d^2x$ (93)

Total momentum: $\mathrm{M} = \displaystyle\int_\Omega h\bar{u} d^2x$ (94)

The objective is thus to choose proper tensor fields $\theta_{\bar{u}}, \theta_h$, and $\theta_\Theta$ for the state variables $\bar{u}, h$, and $\Theta$, respectively, so that $E, \mathcal{M}$, and $\mathrm{M}$ are conserved by the perturbation scheme. Again, it must be emphasized that the conservation law of the perturbation scheme does not directly imply that the same quantities are conserved by the final SPDE.

The domain is 2-dimensional. To conserve mass, the only choice for $\theta_h$ is $\theta_h = h dx^1 \wedge dx^2$, which is a differential $2-$form. It plays the role of density. In order to conserve the momentum, we need the momentum to be a differential 2-form as well. Hence we must choose $\theta_{\bar{u}}$ to be a function (differential 0-form). Therefore, the only choice for $\theta_{\bar{u}}$ is $\theta_{\bar{u}} = \bar{u}$. This choice of $\theta_{\bar{u}}$ and $\theta_h$ implies that $h|\bar{u}|^2$ also corresponds to a 2-form $|\bar{u}|^2\theta_h$. Hence the kinetic energy is automatically conserved by the perturbation scheme. This means that if we want $E$ to be conserved, we must select $\theta_\Theta$ so that $h^2\Theta$ corresponds to a differential $2-$form. Note that $\theta_h$ is already a 2-form. We must thus select $\theta_\Theta$ so that $h\Theta$ corresponds to a function. The only choice for $\theta_\Theta$ is the contravariant tensor $\theta_\Theta = \Theta\frac{\partial}{\partial x^1} \wedge \frac{\partial}{\partial x^2}$. In this case, $h\Theta$ corresponds to the differential $0-$form $\langle\theta_h, \theta_\Theta\rangle = h\Theta$, where $\langle,\rangle$ in this section is the natural pairing of covariant $n-$tensor fields and contravariant $n-$tensor fields.

In sum, we have chosen the following tensor fields:

$\theta_h = h dx^1 \wedge dx^2$ (95)

$\theta_{\bar{u}^j} = \bar{u}^j \quad$ (for $j = 1, 2$) (96)

$\theta_\Theta = \Theta\dfrac{\partial}{\partial x^1} \wedge \dfrac{\partial}{\partial x^2}.$ (97)

For

$T_t(x) = x + a\Delta t + e_i\Delta\eta_i,$ (98)

we have

$T_t^{-1}(x) = x + (-a + e_i^p\partial_{x_p}e_i)\Delta t - e_i\Delta\eta_i.$ (99)

Then $T_t^* \theta_h$, $T_t^* \theta_{\bar{u}}$, and $(T_t^{-1})_* \theta_\Theta$ can be calculated following examples 3.1.3, 3.1.1, and 3.1.5. This further implies $\mathsf{d}_s h, \mathsf{d}_s \bar{u}$, and $\mathsf{d}_s \Theta$, as shown in examples 3.2.2, 3.2.1, and 3.2.4. Note that $T_t^{-1}$ instead of $T_t$ is applied to $\theta_\Theta$ as shown in Eq.(7). Finally, we end up with the following SPDE:

$$
\begin{aligned}
\qquad \mathsf{d}h = &- \nabla(h\bar{u})\mathsf{d}t + \left( h(\partial_{x_p} a^p + \frac{1}{2} J_i) + a^p \partial_{x_p} h + \frac{1}{2} e_i^p e_i^q \partial_{x_p} \partial_{x_q} h + \partial_{x_p} h e_i^p \partial_{x_q} e_i^q \right) \mathsf{d}t \\
&+ (h \partial_{x_p} e_i^p + \partial_{x_p} h e_i^p) \mathsf{d}\eta_i
\end{aligned}
\tag{100}
$$

$$
\begin{aligned}
\mathsf{d}\Theta = &\{-(\bar{u}\cdot\nabla)\Theta - \kappa(h\Theta - h_0\Theta_0)\}\mathsf{d}t \\
&+ \left( \Theta(-\partial_{x_p} a^p + \partial_{x_p}(\partial_{x_q} e_i e_i^q)^p + \frac{1}{2} J_i) + \partial_{x_p}\Theta a^p + \frac{1}{2} e_i^p e_i^q \partial_{x_p} \partial_{x_q}\Theta - \partial_{x_p}\Theta e_i^p \partial_{x_q} e_i^q \right) \mathsf{d}t \\
&- (\Theta \partial_{x_p} e_i^p - \partial_{x_p}\Theta e_i^p) \mathsf{d}\eta_i
\end{aligned}
\tag{101}
$$

$$
\begin{aligned}
\qquad \mathsf{d}\bar{u}^j = &- \{(\bar{u}\cdot\nabla)\bar{u} - f\hat{z}\times\bar{u} - \nabla(h\Theta) + \frac{1}{2} h\nabla\Theta\}^j \mathsf{d}t \\
&+ \left( \partial_{x_p}\bar{u}^j a^p + \frac{1}{2} e_i^p e_i^q \partial_{x_p} \partial_{x_q}\bar{u}^j \right)\mathsf{d}t + \partial_{x_p}\bar{u}^j e_i^p \mathsf{d}\eta_i,
\end{aligned}
\tag{102}
$$

where $J_i = \partial_{x_p} e_i^p \partial_{x_q} e_i^q - \partial_{x_q} e_i^p \partial_{x_p} e_i^q$. And the total mass, total momentum and the total energy shall all be conserved by the perturbation scheme.

## 6 Conclusions

The starting point of this work is to question "how to consistently perturb the location of the state variable?", motivated by Brenier's theorem [Brenier (1991)] which suggests that the difference of two density fields can be represented by a transport map $T$. Noting that optimal transportation has a clean representation in terms of differential $n-$forms, we proposed to perturb the "location" of the state variable $S$, at every forecast time step, by perturbing the corresponding differential $k-$forms $\theta$ by $\theta \leftarrow T_t^* \theta$, where $T_t$ is a random diffeomorphism which deviates from the identity map infinitesimally.

Under this framework, we end up with a stochastic PDE of the state variable $S$ in the form

$$
\mathsf{d}S = f(S)\mathsf{d}t + \mathsf{d}_s S,
\tag{103}
$$

where $f(S)\mathsf{d}t$ is the incremental of $S$ given by the original deterministic system. The term $\mathsf{d}_s S$ is the additional stochastic incremental of $S$ caused by the perturbation scheme.

In this paper, we generalize this scheme to mixed type of tensor fields $\theta$. A key point is indeed to link the state variable 545 $S$ with some tensor field $\theta$. The choice of $\theta$ can then correspond to the conservation laws of certain quantities. We describe in detail how to calculate $T_t^*$ and $T_{t*}$, and present results for several examples corresponding to different choices of $\theta$. We also discussed about the conservation laws for these examples. We emphasize that Brenier's theorem merely serves as the motivation but not the theoretical foundation of the proposed scheme, since the 'optimality' of the displacement vector field needs to be rigorously defined for general tensor fields $\theta$ that are not positive differential $n-$forms.

Interestingly, similarities and differences can be studied between the proposed perturbation scheme and the existing stochastic physical SALT and LU settings Holm (2015); Mémin (2014); Resseguier et al. (2017a). In particular, both SALT and LU equations can be recovered using a prescribed definition of the random diffeomorphism $T_t$ used by the perturbation scheme. For illustration, a stochastic version of the thermal shallow water equation is presented. Compared with SALT and LU settings Holm (2015); Mémin (2014); Resseguier et al. (2017a), the proposed perturbation scheme does not directly rely on the

physics. Hence it is more flexible and can be applied to any PDE. Yet, the proposed derivation also provides interesting means to interpret the operator $\mathcal{L}_{\mathrm{d}x_t - u\mathrm{d}t}$, appearing in the SALT equation. In terms of the optimal transportation, this term represents the infinitesimal forecast error at every forecast time step.

     In order to apply the proposed perturbation scheme to any specific model, the parameters $a$ and $e_i$ must be determined specifically. Hence it is necessary to learn these parameters from existing data, experimental runs, or additional physical

considerations Resseguier et al. (2020, 2021). We anticipate this framework naturally provides a new perspective on how to learn these parameters. Likely, this task will invoke the need of numerical algorithms to estimate the optimal transportation map for general differential $k-$forms or even mixed type of tensor fields. This will be subjects of future investigations.

## Appendix A:  Calculation of $T_t^{-1}$

Suppose that

$$T_t(x) = x + a\Delta t + e_i \Delta\eta_i. \tag{A1}$$

We assume that $T_t^{-1}$ has the following form of expression:

$$T_t^{-1}(x) = x + z\Delta t + b_i \Delta\eta_i. \tag{A2}$$

Our goal is to find $z$ and $b_i$. Then we have

$$
\begin{aligned}
x =& T_t(T_t^{-1}(x)) = T_t(x + z\Delta t + b_i \Delta\eta_i) \\
=& x + z\Delta t + b_i \Delta\eta_i + a\Big|_{x+z\Delta t+b_i\Delta\eta_i}\Delta t + e_i\Big|_{x+z\Delta t+b_i\Delta\eta_i}\Delta\eta_i
\end{aligned}
\tag{A3}
$$

Similar to the derivation in section (3.1), we apply Taylor expansion and Itô's lemma, and drop the terms of higher-order infinitesimal:

$$
\begin{aligned}
a\Big|_{x+a\Delta t+b_i\Delta\eta_i}\Delta t =& a\Big|_x \Delta t + o(\Delta t) \\
e_i\Big|_{x+z\Delta t+b_i\Delta\eta_i}\Delta\eta_i =& e_i\big|_x \Delta\eta_i + e_{ip}b_i^p\Big|_x \Delta t + o(\Delta t).
\end{aligned}
\tag{A4}
$$

Therefore

$$x = T_t(T_t^{-1}(x)) = x + (z + a + e_{ip}b_i^p)\Delta t + (b_i + e_i)\Delta\eta_i + o(\Delta t). \tag{A5}$$

This implies that

$$b_i + e_i = 0 \tag{A6}$$

$$z + a + e_{ip}b_i^p = 0 \tag{A7}$$

Therefore

$$b_i = -e_i \tag{A8}$$

$$z = -a + e_{ip}e_i^p, \tag{A9}$$

or equivalently,

$$T_t^{-1}(x) = x + (-a + e_{ip}e_i^p)\Delta t - e_i\Delta\eta_i \tag{A10}$$

## Appendix B:  Derivation of $T_t^*\theta$

Given coordinates $(x^1, ..., x^n)$, when $\theta$ is a differential $k-$form, it can be written as

$$\theta = \sum_{i_1 < ... < i_k} f^{i_1, ..., i_k} dx^{i_1} \wedge \cdots \wedge dx^{i_k}. \tag{B1}$$

Since $T_t^*$ is linear, we may assume that

$$\theta = f dx^{i_1} \wedge \cdots \wedge dx^{i_k} \tag{B2}$$

for some $1 \le i_1 < \cdots < i_k \le n$. Let $T_t(x) = (T_t^1(x), ..., T_t^n(x))$, then

$$(T_t^*\theta)(x) = f(T_t(x)) dT_t^{i_1} \wedge \cdots \wedge dT_t^{i_k}. \tag{B3}$$

We calculate $f(T_t(x))$ and $dT_t^{i_1} \wedge \cdots \wedge dT_t^{i_k}$ separately. We denote $\Delta x = T_t(x) - x = a\Delta t + e_i\Delta\eta_i$, and $H_f$ the Hessian matrix of $f$. At a given time $t$, $f$ is assumed independent from the noise $\Delta\eta_i(t)$. Then

$$f(T_t(x)) = f(x + \Delta x) = f(x) + \langle \nabla f, \Delta x \rangle + \frac{1}{2}(\Delta x)^\top H_f \Delta x + o((\Delta x)^2) \tag{B4}$$

$$= f(x) + \langle \nabla f, a\Delta t + e_i\Delta\eta_i \rangle + \frac{1}{2}e_i^\top H_f e_i(\Delta\eta_i)^2 \tag{B5}$$

$$+ \mathcal{O}((\Delta t)^2) + \mathcal{O}(\Delta t\Delta\eta_i) + o((\Delta t)^2) + o((\Delta\eta_i)^2) + o(\Delta t\Delta\eta_i) \tag{B6}$$

According to Itô's lemma $d\eta d\eta = dt$, and we can replace $(\Delta\eta_i)^2$ with $\Delta t$. Hence

$$f(T_t(x)) = f(x) + \langle \nabla f, a \rangle \Delta t + \langle \nabla f, e_i \rangle \Delta\eta_i + \frac{1}{2}e_i^\top H_f e_i \Delta t + o(\Delta t) \tag{B7}$$

$$= f(x) + \left( \langle \nabla f, a \rangle + \frac{1}{2}e_i H_f e_i \right)\Delta t + \langle \nabla f, e_i \rangle \Delta\eta_i + o(\Delta t). \tag{B8}$$

Next,

$$T_t^*(dx^{i_1} \wedge \cdots \wedge dx^{i_k}) = dT_t^{i_1} \wedge \cdots \wedge dT_t^{i_k}$$

$$= (dx^{i_1} + da^{i_1}\Delta t + de_i^{i_1}\Delta\eta_i) \wedge \cdots \wedge (dx^{i_k} + da^{i_k}\Delta t + de_i^{i_k}\Delta\eta_i). \tag{B9}$$

Note that $da^{i_j}$ and $de_i^{i_j}$ refer to the spatial differentiation. Again, we apply the "discrete version" of Itô's rule $(\Delta\eta_i)^2 = \Delta t$, and collect all the terms of order $\mathcal{O}(\Delta t)$ and $\mathcal{O}(\Delta\eta_i)$:

$$T_t^*(dx^{i_1} \wedge \cdots \wedge dx^{i_k}) = dx^{i_1} \wedge \cdots \wedge dx^{i_k} + \Big(\sum_{s=1}^{k} dx^{i_1} \wedge \cdots \wedge da^{i_s} \wedge \cdots \wedge dx^{i_k}\Big)\Delta t$$

$$+ \Big(\sum_{s=1}^{k} dx^{i_1} \wedge \cdots \wedge de_i^{i_s} \wedge \cdots \wedge dx^{i_k}\Big)\Delta\eta_i$$

$$+ \Big(\sum_{s<r} dx^{i_1} \wedge \cdots \wedge de_i^{i_s} \wedge \cdots \wedge de_i^{i_r} \wedge \cdots \wedge dx^{i_k}\Big)\Delta t$$

$$+ o(\Delta t) \tag{B10}$$

According to the chain rule, $da^{i_s} = \partial_{x^j} a^{i_s} dx^j$, $de_i^{i_s} = \partial_{x^j} e_i^{i_s} dx^j$. Note that $\partial_{x^j} e_i^{i_s}$ refers to the $i_s$-th component of $\partial_{x^j} e_i$,
where $\partial_{x^j} e_i = \frac{\partial e_i}{\partial x^j}$ and $e_i(x) \in \mathbb{R}^n$ is the $i$−th basis vector field of $T_t$. Hence

$$T_t^*(dx^{i_1} \wedge \cdots \wedge dx^{i_k})$$

$$= dx^{i_1} \wedge \cdots \wedge dx^{i_k} + \Big(\sum_{s=1}^{k} \partial_{x^j} a^{i_s} dx^{i_1} \wedge \cdots \wedge dx^j \wedge \cdots \wedge dx^{i_k}\Big)\Delta t$$

$$+ \Big(\sum_{s=1}^{k} \partial_{x^j} e_i^{i_s} dx^{i_1} \wedge \cdots \wedge dx^j \wedge \cdots \wedge dx^{i_k}\Big)\Delta\eta_i$$

$$+ \Big(\sum_{s<r} \partial_{x^j} e_i^{i_s} \partial_{x^l} e_i^{i_r} dx^{i_1} \wedge \cdots \wedge dx^j \wedge \cdots \wedge dx^l \wedge \cdots \wedge dx^{i_k}\Big)\Delta t$$

$$+ o(\Delta t) \tag{B11}$$

Combining Eqs.(B8) and (B11), with application of Itô's lemma, all terms of order $o(\Delta t)$ are then removed, to obtain

$$
\begin{aligned}
T_t^* \theta =& f(T_t(x)) T_t^* (dx^{i_1} \wedge \cdots \wedge dx^{i_k}) \\
=& \theta + \Big\{ \big( \langle \nabla f, a \rangle + \frac{1}{2} e_i^\top H_f e_i \big) dx^{i_1} \wedge \cdots \wedge dx^{i_n} \\
& + \sum_{s=1}^{k} f \partial_{x^j} a^{i_s} dx^{i_1} \wedge \dots dx^j \wedge \cdots \wedge dx^{i_k} \\
& + \big( \sum_{s<r} f \partial_{x^j} e_i^{i_s} \partial_{x^l} e_i^{i_r} dx^{i_1} \wedge \cdots \wedge dx^j \wedge \cdots \wedge dx^l \wedge \cdots \wedge dx^{i_k} \big) \\
& + \big( \sum_{s=1}^{k} \langle \nabla f, e_i \rangle \partial_{x^j} e_i^{i_s} dx^{i_1} \wedge \cdots \wedge dx^j \wedge \cdots \wedge dx^{i_k} \big) \Big\} \Delta t \\
& + \Big\{ \langle \nabla f, e_i \rangle dx^{i_1} \wedge \cdots \wedge dx^{i_k} + \sum_{s=1}^{k} f \partial_{x^j} e_i^{i_s} dx^{i_1} \wedge \cdots \wedge dx^j \wedge \cdots \wedge dx^{i_k} \Big\} \Delta \eta_i \\
& + o(\Delta t).
\end{aligned}
\tag{B12}
$$

To simplify Eq.(B12), wedge algebra is applied and the high-order infinitesimal $o(\Delta t)$ is ignored. Accordingly, $T_t^* \theta$ is more compactly written as

$$
T_t^* \theta = \theta + \mathcal{M}(\theta) \Delta t + \mathcal{N}_i(\theta) \Delta \eta_i,
\tag{B13}
$$

for some differential $k-$forms $\mathcal{M}(\theta)$ and $\mathcal{N}_i(\theta)$.

*Author contributions.* The first author contributed the main idea and mathematical derivation, and wrote the first draft of the paper. The second author contributed all the physical interpretations of the derived equations, a much more detailed introduction, and a more detailed comparison of the proposed scheme with the LU equation. The third author contributed his insight and enthusiasm on this problem and worked on the writing of the draft. This work is done based on the fruitful discussion of all the three authors since the very early stage.

*Competing interests.* The authors declare no competing interests.

*Acknowledgements.* The authors would like to express their gratitude towards Wei Pan, Darryl Holm, Dan Crisan, Long Li, and Etienne Mémin for their patient explanation and insightful discussion. The research of YZ was supported by the ANR Melody project when he was a postdoc at Ifremer. The research VR is supported by the company SCALIAN DS and by France Relance through the MORAANE project. The research BC is supported by ERC EU SYNERGY Project No. 856408-STUOD, and the the support of the ANR Melody project.

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
