# Peer review of "Physically Constrained Covariance Inflation from Location Uncertainty"

_EGUsphere, 2023_

## Referee Comment (RC1)

The authors provide a general framework for stochastic perturbations of PDEs dependent on a chosen association of the components of the PDE with a differential form, and some further parameters. They show that with the correct choice, one may deduce the SALT and LU perturbations out of this framework which makes this an interesting result worthy of publication.

I would suggest a minor improvement on the presentation is in order before publication. In particular, I believe that a compact summary of the procedure making explicit which choices are to be made by the user, what guides those choices, and the determined steps from these choices, should be included in the introduction. This is partly done in Lines 77 to 80, but it should be made clear in this bullet point summary that (at least as I understand) $\theta$ is non-uniquely chosen, $T$ is non-uniquely chosen, and the rest of the procedure is determined from these choices. A comment on how these choices can be made, as in what needs to be followed in making these choices, and what guides the choices, should be included in this summary. Moreover there should be a similar discussion in the introduction as to how the SALT and LU schemes are recovered in this setting; is there a natural way to choose $\theta$ and $T$ based on the principles of these schemes, which genuinely gives an alternative derivation of the SALT and LU SPDEs, or are you just equating coefficients knowing what outcome you want to achieve and selecting them accordingly? This difference is significant in the purpose of the paper.

Some specific points are given below:

1. In line 2/3, is 'asserting that the difference...' the content of Brenier's Theorem, or is this something which you are implementing in light of Brenier's Theorem? If the former then you should remove the comma at the end of line 2, if the latter then you should give a brief comment before that comma on what Brenier's Theorem is.

2. Similarly in line 4 I assume that 'perturbations are demonstrated...' is what you achieve in the paper, though this should be made more clear what as it could be understood to be part of Brenier's work. Thus you should write 'we demonstrate that perturbations...'.

3. Line 4 should read 'an SPDE'.

4. Line 5/6, what is the reference to 'Resseguier et al' here? If this is another setting like LU and SALT then 'both' should not be used as it applies to all three. If not, then what is this relevance of this paper here?

5. Of course in Line 9 the copyright statement needs to be presented.

6. Line 12 should read 'academic studies'

7. Line 13 'estimate of a state variable'

8. Line 20 a reference should be given for the 'curse of dimensionality'.

9. Line 26 'addition of noise'

10. Line 28 'we refer'

11. Line 36 'multiplicative noise'

12. Line 43/44, 'slowly-varying' and 'fast-varying' should be consistent e.g. replace 'slowly' with 'slow' or 'fast' with 'quickly'.

13. Line 46 'homogenization methods... may also lead to violation of energy conservation', can you comment on why this is the case, for example what the mentioned workarounds are protecting against?

14. Line 52 'another'

15. Line 88 'In summary' and 'provides with the perspeective that' should be reworded.

16. Line 95 'we reserve this for future study'.

17. Line 101 'A final conclusion and discussion is given...'

18. Line 105 'if p is shown in both upper and lower indices'

19. Line 147 'an SPDE'

20. Line 162 'a detailed definition'

21. Line 162 'a Taylor Expansion and Itô's Lemma can be used'

22. Line 168 'a Taylor Expansion'

23. Line 195 What is this $\mathcal{F}$? What is the space of $f$ and space of $\theta$? There needs to be more precision about what is happening here, and how this $\mathcal{F}$ is known to exist or how it is chosen/can be chosen.

24. Line 203 should read 'As the physical PDE (20) is deterministic'

25. Lines 231, 233 should end in full stops, and this implies in general for equations which are followed by a new sentence (starting with a capital letter).

26. Line 256 'Again an advection...'. Also 'for an n-form'

27. Line 259 'provides an'

28. Line 271 'is remeniscent of'

29. Line 273 'from the cross...'. Also It$\bar{o}$ should be replaced by Itô.

30. Line 284 needs to be reworded

31. Line 347 I now understand the reference to Resseguier to be as a secondary reference for Location Uncertainty, though this should be made clear.

32. Line 356 and throughout should say 'Itô'.

33. Line 388 'already outlined'

34. Line 393, 'terms, the LU equation'

35. Line 428 'used as an advection velocity in the SALT'

36. Line 533 'needs to be'

---

## Author Comment (AC2)

**Physically Constrained Covariance Inflation from Location Uncertainty**
**Revision notes**

Yicun Zhen* [1], Valentin Resseguier[2], and Bertrand Chapron[3]

[1] *Coloege of Oceanography, Hohai University, China*
[2] *LAB SCALIAN DS, Rennes, France*
[3] *Laboratoire d'Océanographie Physique et Spatiale, Ifremer, Plouzaé, France*

Dear Editors,

We express our gratitude for the time and effort dedicated to the reviewing of our submitted manuscript. We worked diligently to address all the concerns raised by the referees. Below we provide our detailed response to their comments. **We also include another copy of the PDF file of the manuscript on which all the changes have been marked.** The marked copy can help the reviewer to see the difference of the manuscripts before and after revision. We hope that the applied revisions are to the satisfaction of the editors.

Kind regards,

Yicun Zhen, Valentin Resseguier, Bertrand Chapron

**Manuscript information**

**Number:** Preprint egusphere-2023-416

**Title:** "Physically Constrained Covariance Inflation from Location Uncertainty"

**Authors:** Yicun Zhen, Valentin Resseguier, Bertrand Chapron

**Submitted to:** Nonlinear Processes in Geophysics
* * *
*Corresponding author
✉ Coloege of Oceanography, Hohai University, China
✐ zhenyicun@protonmail.com

**Reviewer 1**

**Comment I (Reviewer 1):**

*The authors provide a general framework for stochastic perturbations of PDEs dependent on a chosen association of the components of the PDE with a differential form, and some further parameters. They show that with the correct choice, one may deduce the SALT and LU perturbations out of this framework which makes this an interesting result worthy of publication. I would suggest a minor improvement on the presentation is in order before publication. In particular, I believe that a compact summary of the procedure making explicit which choices are to be made by the user, what guides those choices, and the determined steps from these choices, should be included in the introduction. This is partly done in Lines 77 to 80, but it should be made clear in this bullet point summary that (at least as I understand) θ is non-uniquely chosen, T is non-uniquely chosen, and the rest of the procedure is determined from these choices. A comment on how these choices can be made, as in what needs to be followed in making these choices, and what guides the choices, should be included in this summary. Moreover there should be a similar discussion in the introduction as to how the SALT and LU schemes are recovered in this setting; is there a natural way to choose θ and T based on the principles of these schemes, which genuinely gives an alternative derivation of the SALT and LU SPDEs, or are you just equating coefficients knowing what outcome you want to achieve and selecting them accordingly? This difference is significant in the purpose of the paper.*

*Some specific points are given below:*

**Reply:** *We appreciate the reviewer's careful reading and his patience in pointing out all the grammatical mistakes. Our point-by-point response can be found in the following.*

**1**, *In line 2/3, is 'asserting that the difference...' the content of Brenier's Theorem, or is this something which you are implementing in light of Brenier's Theorem? If the former then you should remove the comma at the end of line 2, if the latter then you should give a brief comment before that comma on what Brenier's Theorem is.*

**Reply**: *It is the former. We have deleted the comma.*

**2**. *Similarly in line 4 I assume that 'perturbations are demonstrated...' is what you achieve in the paper, though this should be made more clear what as it could be understood to be part of Brenier's work. Thus you should write 'we demonstrate that perturbations...'.*

**Reply**: *We have made the changes as the reviewer requested.*

**3**, *Line 4 should read 'an SPDE'.*

**Reply**: *Thanks the reviewer for pointing it out. It has been fixed.*

**4**. *Line 5/6, what is the reference to 'Resseguier et. al' here? If this is another setting like LU and SALT then 'both' should not be used as it applies to all three. If not, then what is this relevance of this paper here?*

**Reply**: *Mémin (2014) developed the initial idea of location uncertainty equation. But this original paper additionally assumed strong smoothness properties of the solution, to eventually remove the noises terms of the original Navier-Stokes equations under location uncertainty. Since Resseguier et al. (2017a) (erroneously cited as Resseguier et al. (2016) in the original manuscript) and Resseguier et al. (2017b), this misleading assumption was removed, in order to keep the important noise terms. Since then, the original deterministic LU Navier-Stokes equations have been referred to as pseudo-stochastic Navier-Stokes equations (Resseguier et al., 2021). We list Resseguier et al. (2017a) in the abstract to guide the readers to the LU equations of interest for us, i.e. the equations involving noises.*

**5**. *Of course in Line 9 the copyright statement needs to be presented.*

**Reply**: *Thanks the reviewer for reminding us. We have removed this section since there is no copyright transfer.*

**6**. *Line 12 should read 'academic studies'*

**Reply**: *fixed.*

**7**. *Line 13 'estimate of a state variable'*

**Reply**: *fixed.*

**8**. *Line 20 a reference should be given for the 'curse of dimensionality'.*

**Reply**: *A reference has been added.*

**9**. *Line 26 'addition of noise'*

**Reply**: *fixed.*

**10**. *Line 28 'we refer'*

**Reply**: *fixed.*

**11**. *Line 36 'multiplicative noise'*

**Reply**: *fixed.*

**12**. *Line 43/44, 'slowly-varying' and 'fast-varying' should be consistent e.g. replace 'slowly' with 'slow' or 'fast' with 'quickly'.*

**Reply**: *fixed.*

**Comment II (Reviewer 1):**

**13**. *Line 46 'homogenization methods... may also lead to violation of energy conservation', can you comment on why this is the case, for example what the mentioned workarounds are protecting against?*

**Reply**: *Homogenization methods may lead to violation of energy conservation for several reasons. First, even if the energy of the whole slow-fast system is conserved, the energy of the resolved slow component is not in general since it should exchange energy with the unresolved fast component. Secondly, the energy fluxes (triads) between the modes (say the Fourier modes) of the slow resolved component are often functions of the fast unresolved component. Without properly resolving the latest, triads can be strongly erroneous. Thirdly, prior to homogenization, authors often simply the dynamics of the fast component (e.g. Majda et al., 1999; Frank and Gottwald, 2013). This approximation often breaks the energy conservation. The workarounds enable energy conservation anyhow. We noticed that we cited the wrong paper here: Gottwald and Melbourne (2013) instead of Frank and Gottwald (2013). We corrected that.*

**14**. *Line 52 'another'*

**Reply**: *fixed.*

**15**. *Line 88 'In summary' and 'provides with the perspeective that' should be reworded.*

**Reply**: *We have rewritten the sentence.*

**16**. *Line 95 'we reserve this for future study'.*

**Reply**: *fixed.*

**17**. *Line 101 'A final conclusion and discussion is given...'*

**Reply**: *fixed.*

**18**. *Line 105 'if p is shown in both upper and lower indices'*

**Reply**: *fixed.*

**19**. *Line 147 'an SPDE'*

**Reply**: *fixed.*

**20**. *Line 162 'a detailed definition'*

**Reply**: *fixed.*

**21**. *Line 162 'a Taylor Expansion and Itô's Lemma can be used'*

**Reply**: *fixed.*

**22**. *Line 168 'a Taylor Expansion'*

**Reply**: *fixed.*

**23**. *Line 195 What is this F? What is the space of f and space of θ? There needs to be more precision about what is happening here, and how this F is known to exist or how it is chosen/can be chosen.*

**Reply**: *the corresponding map $\mathcal{F}$ is obvious once the type of tensor field has been chosen. We have added some words exaplaining this.*

**24**. *Line 203 should read 'As the physical PDE (20) is deterministic'*

**Reply**: *fixed.*

**25**. *Lines 231, 233 should end in full stops, and this implies in general for equations which are followed by a new sentence (starting with a capital letter).*

**Reply**: *fixed.*

**26**. *Line 256 'Again an advection...'. Also 'for an n-form'*

**Reply**: *fixed.*

**27**. *Line 259 'provides an'*

**Reply**: *fixed.*

**28**. *Line 271 'is remeniscent of'*

**Reply**: *fixed.*

**29**. *Line 273 'from the cross...'. Also Ito should be replaced by Itô.*

**Reply**: *fixed.*

**Comment III (Reviewer 1):**

**30**. *Line 284 needs to be reworded*

**Reply**: *fixed.*

**31**. *Line 347 I now understand the reference to Resseguier to be as a secondary reference for Location Uncertainty, though this should be made clear.*

**Reply**: *The pseudo-stochastic LU Navier-Stokes of Mémin (2014) is a deterministic PDE and hence cannot be recovered by our stochastic scheme, whereas we can recover the stochastic LU Navier-Stokes equations, originated from Resseguier et al. (2017a). We will try to make this clear at the beginning of section 4 in the revised manuscript.*

**32**. *Line 356 and throughout should say 'Itô'.*

**Reply**: *fixed.*

**33**. *Line 388 'already outlined'*

**Reply**: *fixed.*

**34**. *Line 393, 'terms, the LU equation'*

**Reply**: *fixed.*

**35**. *Line 428 'used as an advection velocity in the SALT'*

**Reply**: *fixed.*

**36**. *Line 533 'needs to be'*

**Reply**: *fixed.*

**Reviewer 2**

**Comment IV (Reviewer 2):**

*This paper improves on previous methods for "covariance inflation" to arrive at perturbations of abstract tensor quantities rather than positive definite scalar quantities. This requires the application of optimal transport theory and Brenier's answer to it. This represents an important incremental step in aligning and rectifying the different approaches to including Location Uncertainty (e.g., LU and SALT approaches nicely reviewed in the introduction). I have only minor technical suggestions.*

*Some of the math developments go by a bit quickly without reference to the underlying concepts. It would strengthen the manuscript to include them, e.g.,*

**Reply:** *We appreciate the reviewer's careful reading. Our point-by-point response can be found in the following.*

- *Section 3.1–it would be nice to name the different forms in (13)-(16) for later reference or just to ground the notation.*

  **Reply**: *we appreciate the reviewer's suggestion on naming the different differential forms. We understand that the reviewers is suggesting us giving guidance on how to link the physical quantities to the tensor fields. We have added a short explanation on this point right after Eq.(17). But we think it is hard to give names to the differential forms. Because the choice of differential forms could differ case by case. As we already seen, the LU equation treats the velocity components as functions while the SALT equation treats the velocity as differential $1-$forms. Usually when a name is given, its interpretation is fixed. We think it is better to give the reader some room to have their own interpretation of the physical quantities.*

- *Section 3.2–Just after (3.2) it would be nice to link to a reference on CFL conditions there.*

  **Reply**: *We added the Remark 5 dealing with this point.*

- *Just after (75), please state in words what those equations (29, 75) do so the reader doesn't have to scan backwards.*

  **Reply**: *we have added a sentence reminding the readers what Eq.(29) is.*

- *Typo: just before (65) a reference requires parentheses. Same issue on Line 520.*

  **Reply**: *fixed.*

**References**

Frank, J. E. and Gottwald, G. A.: Stochastic homogenization for an energy conserving multi-scale toy model of the atmosphere, Physica D: Nonlinear Phenomena, 254, 46–56, 2013.

Gottwald, G. and Melbourne, I.: Homogenization for deterministic maps and multiplicative noise, Proceedings of the Royal Society of London A: Mathematical, Physical and Engineering Sciences, 469, 2013.

Majda, A. J., Timofeyev, I., and VandenEijnden, E.: Models for stochastic climate prediction., Proceedings of the National Academy of Sciences of the United States of America, 96 26, 14 687–91, 1999.

Mémin, E.: Fluid flow dynamics under location uncertainty, Geophysical & Astrophysical Fluid Dynamics, 108, 119–146, 2014.

Resseguier, V., Mémin, E., and Chapron, B.: Geophysical flows under location uncertainty, Part I Random transport and general models, Geophysical & Astrophysical Fluid Dynamics, 111, 149–176, 2017a.

Resseguier, V., Mémin, E., and Chapron, B.: Geophysical flows under location uncertainty, Part II Quasi-geostrophy and efficient ensemble spreading, Geophysical & Astrophysical Fluid Dynamics, 111, 177–208, 2017b.

Resseguier, V., Li, L., Jouan, G., Dérian, P., Mémin, E., and Chapron, B.: New trends in ensemble forecast strategy: uncertainty quantification for coarse-grid computational fluid dynamics, Archives of Computational Methods in Engineering, 28, 215–261, 2021.